# Single-cell transcriptomics-informed induced pluripotent stem cells differentiation to tenogenic lineage

Angela Papalamprou[1,2,3], Victoria Yu[1,2,3], Wensen Jiang[1,2,3], Julia Sheyn[1,2,3], Tina Stefanovic[1,2,3], Angel Chen[1,2,3], Chloe Castaneda[1,2,3], Melissa Chavez[1,2,3], Dmitriy Sheyn[1,2,3,4,5]*

[1]Orthopaedic Stem Cell Research Laboratory, Cedars-Sinai Medical Center, Los Angeles, United States; [2]Board of Governors Regenerative Medicine Institute, Cedars-Sinai Medical Center, Los Angeles, United States; [3]Department of Biomedical Sciences, Cedars-Sinai Medical Center, Los Angeles, United States; [4]Department of Orthopedics, Cedars-Sinai Medical Center, Los Angeles, United States; [5]Department of Surgery, Cedars-Sinai Medical Center, Los Angeles, United States

## eLife Assessment

The authors established a **useful** syndetome differentiation protocol from human induced pluripotent stem cells, guided by single-cell transcriptomic analysis. Their findings could significantly impact the field, particularly for patients needing tendon cell therapy. However, the evidence presented is currently **incomplete**, as the authors did not yet test the applicability of their protocol across multiple human induced pluripotent stem cell lines.

*For correspondence:
Dmitriy.Sheyn@csmc.edu

Competing interest: The authors declare that no competing interests exist.

**Abstract** During vertebrate embryogenesis, axial tendons develop from the paraxial mesoderm and differentiate through specific developmental stages to reach the syndetome stage. While the main roles of signaling pathways in the earlier stages of differentiation have been well established, pathway nuances in syndetome specification from the sclerotome stage have yet to be explored. Here, stepwise differentiation of human induced pluripotent stem cells to the syndetome stage is shown, using chemically defined media and small molecules that were modified based on single-cell RNA-sequencing and pathway analysis. A significant population of branching off-target cells differentiating toward a neural phenotype overexpressing Wnt was identified. Further transcriptomics post-addition of a WNT inhibitor at the somite stage and onwards revealed not only total removal of the neural off-target cells, but also increased syndetome induction efficiency. Fine-tuning tendon differentiation in vitro is essential to address the current challenges in developing a successful cell-based tendon therapy.

## Introduction

Tendons are necessary to support body movement by transferring forces between muscle and bone. Unfortunately, tendon injuries are quite prevalent in athletes and in the aging population, comprising 45% of musculoskeletal consultations in the U.S. alone (*Yang et al., 2013*). Tendons are structurally complex tissues, and their low vascularity and cellularity are contributing factors to their poor regenerative capacity (*Liu et al., 2017*). The current standard of treatments for severe tendon injury includes conservative approaches and/or surgical intervention (*Yang et al., 2013*). However, the resulting prolonged rehabilitation, muscle weakness, and high re-injury rates dramatically limit patient

outcomes (*Webster et al., 2014*). Development of a stem cell-mediated solution could improve patient outcomes, yet current approaches fall short in terms of tissue biomechanical performance and tissue organization (*Puetzer et al., 2021*; *Cai et al., 2020*; *Ho et al., 2014*).

Cell-based therapies have shown potential for tendon repair (*Abbah et al., 2014*). Autologous tendon progenitors and mesenchymal stromal cells (MSCs) have been explored as potential cell sources; however, their limitations for clinical application are associated with their heterogeneity and the need for in vitro expansion for obtaining clinically relevant cell numbers. Cell expansion in vitro can result in phenotypic drift and subsequent functional loss, in addition to low proliferative ability (*Jo et al., 2019*). Utilizing tenocytes differentiated from induced pluripotent stem cells (iPSCs) holds promise due to their unparalleled developmental plasticity, unlimited self-renewal capacity, and the potential scalability for an off-the-shelf cell source application. Well-established protocols have been developed for the differentiation from pluripotent stem cells to different musculoskeletal cell types, including chondrocytes and osteoblasts (*Wang et al., 2013*; *Kanke et al., 2014*; *Yamashita et al., 2015*; *Wu et al., 2021*). Recent studies have successfully differentiated tenocytes using mouse iPSCs and embryonic stem cells (*Komura et al., 2020*; *Nakajima and Ikeya, 2021*; *Kaji et al., 2021*; *Yoshimoto et al., 2022*). Though, recent work has shown a divergence in developmental processes between mouse and human embryos and subsequent differences in developmental cues on tenogenic differentiation between species (*Donderwinkel et al., 2022*; *Brown et al., 2015*; *Havis et al., 2014*; *Havis et al., 2016*). This highlights the importance of researching the developmental cues of human cells. Furthermore, other studies have either not reported induction efficiency or showed limited syndetome-like/tenocyte induction (*Komura et al., 2020*; *Dale et al., 2018*). Deriving a homogenous population of tenocytes from iPSCs with high efficiency continues to be a challenge, largely in part to the limited understanding of their developmental origins and differentiation path from multipotential precursors.

In vertebrate embryogenesis, tendons arise from mesodermal cells of different origin. Axial, limb, and cranial tendons originate from paraxial mesoderm, lateral plate mesoderm, or neural crest (NC), respectively. Despite differential cell and tissue interactions between the three regions, the major molecular regulators of tendon differentiation are shared between all three (*Schweitzer et al., 2010*; *Huang et al., 2015*). BMP, TGFβ, Activin/Nodal, FGF, and WNT signaling pathways regulate mesoderm induction from pluripotent cells. Along the mediolateral axis of the embryo, the balance between BMP and WNT signaling gradients specifies mesodermal subtypes (*Tani et al., 2020*; *Loh et al., 2016*). The paraxial or presomitic mesoderm (PSM) is derived from neuromesodermal progenitors (NMPs) that are bipotential and able to differentiate into ectodermal and mesodermal lineages (*Tani et al., 2020*). Recent in vitro studies demonstrated that during mesoderm specification, WNT controls the allocation to PSM and represses lateral plate mesoderm (*Loh et al., 2016*; *Nakajima et al., 2018*). Somites (SM) are derived from the anterior PSM following dynamic morphogenetic cyclic signaling, involving Notch, WNT, and FGF (*Tani et al., 2020*). Somite progenitors are multipotential and are further specified by signaling molecules from the surrounding tissues. Sonic hedgehog (SHH) is secreted from the notochord and the neural tube, and it has been demonstrated as a crucial signaling modulator in sclerotome (SCL) specification of somites combined with low levels of WNT and BMP (*Tani et al., 2020*; *Loh et al., 2016*). Murine and avian embryonic development studies concluded that combined SHH activation and BMP inhibition may be sufficient for SCL induction (*Tani et al., 2020*). However, it was later shown in vitro that WNT directly antagonizes SHH and induces somites to dermomyotome (*Loh et al., 2016*). Since SCL is in contact with different cell populations and thus diverging signals, it gives rise to different cell populations along the three major patterning axes. SCL develops dorsally into the syndetome (SYN), the precursor of tendons of the body trunk, which is marked by the expression of the transcription factor scleraxis (SCX) (*Tani et al., 2020*). Recent studies have indicated that FGF, BMP, and TGFβ are orchestrating syndetome differentiation from sclerotome (*Kaji et al., 2021*; *Yoshimoto et al., 2022*; *Nakajima et al., 2021*). A few groups have differentiated mouse iPSCs (*Kaji et al., 2021*; *Yoshimoto et al., 2022*; *Nakajima et al., 2021*) and human iPSCs (*Nakajima et al., 2021*) to SYN through SCL and have pinpointed the primary role of BMP and TGFβ signaling. However, the nuances of WNT signaling in SYN specification have yet to be explored.

In this study, we sought to fine-tune induction to SYN by further elucidating the signaling pathways involved with tenocyte formation using single-cell transcriptomics. SYN was induced from GMP-ready

human iPSCs by stepwise differentiation through PSM-derived SM cells using chemically defined media. At every stage of the induction, we monitored known markers for each developmental step and conducted further characterization with single-cell RNA sequencing (scRNA-seq) and immunostaining. ScRNA-seq analysis revealed off-target differentiation toward a neural phenotype, with WNT family members being identified as crucial to the generation of neural by-products. We hypothesized that WNT signaling inhibition would block cell fate bifurcation to 'unwanted' states and drive induction toward a single path. Informed by single-cell transcriptomics, we added the WNT inhibitor Wnt-C59 to the later stages of the differentiation, which improved final induction efficiency of the differentiated tendon progenitor population.

## Results

### Stepwise induction of iPSCs to syndetome-like cells using chemically defined media and small molecules in vitro

Human iPSCs were seeded at different densities and induced to PSM for 4 days with the GSK3 inhibitor (GSK3i) CHIR99021 leading to WNT pathway activation, combined with BMP and TGFβ inhibition and concurrent FGF activation (*Figure 1A*). Gene expression analysis, immunofluorescent (IF) staining, and flow cytometry for DLL1 levels were used to determine induction efficiency and establish the starting seeding density (*Figure 1—figure supplement 1*). A homogeneous population of DLL1+ cells was generated after 4 days when cells were seeded at ~40–50 aggregates/cm$^2$. Specifically, 96.2% DLL1$^+$ cells were generated at day 4 (*Figure 1—figure supplement 1*). In contrast, DLL1$^+$ cells were shown to be 43.6% less (52.6% DLL1$^+$ cells) when iPSCs were seeded at 2x concentration.

Cells exhibited distinct morphologies at different stages of chemical induction (*Figure 1B*). On day 4, iPSC colonies displayed contraction, resulting in differentiating cells leaving the colonies at the periphery. By day 8, cells were completely dissociated from the colonies. On day 11, colonies looked contracted and regressed, and were surrounded by spindly cells. Lastly, by the end of the differentiation period, fibroblast-like cells had grown in size. However, during SYN induction we noticed a gradual attrition of cells, resulting in considerably lower cell numbers at the end of the process.

To further characterize cell differentiation at each stage, the expression of PSM, SM, and SCL stage cell markers was examined with RT-qPCR (*Figure 1C–F*) and IF (*Figure 2A–G*). Additionally, markers associated with tendon progenitors and tenocytes were examined as an indication of SYN differentiation (*Figures 1G and 2H, I*). Pluripotency markers (OCT4, NANOG) were significantly downregulated at PSM compared to iPSC and were not detected at subsequent stages (*Figure 1C*). PSM marker genes DLL1, TBX6, WNT3A, and MSGN1 (*Tani et al., 2020*; *Loh et al., 2016*) were significantly upregulated at days 3 and 4 of GSK3i treatment and downregulated after 4 more days of TGFβ inhibition and WNT activation of DLL1$^+$ cells (*Figure 1D*). Mesodermal T-box transcription factors TBX1 and TBX6 (*Gentsch et al., 2017*) were detected at PSM stage (*Figure 2C*) but not at SM stage (*Figure 2C, D*). SM markers MEOX1 and PAX3 (*Tani et al., 2020*; *Loh et al., 2016*) were not expressed at PSM stage (*Figures 1E and 2C–E*). Following 3 days of BMP inhibition and SHH pathway activation of SM stage cells, SCL markers (NKX3.2, PAX1, and PAX9) (*Tani et al., 2020*; *Loh et al., 2016*) were significantly upregulated at day 11 (*Figure 1F*). PAX1 and PAX9 were observed in the entire cell population at SCL (*Figure 2G*). PDGFRA and TPPP3 (*Harvey et al., 2019*) markers were significantly upregulated at the SYN stage (*Figure 1G*). Interestingly, SCX, a prominent tenogenic transcription factor (*Shukunami et al., 2018*) was significantly downregulated at the SCL stage compared to iPSC, but upregulated during the differentiation from SCL to SYN. Further, COL1A1, COL3A1, and TNMD (*Liu et al., 2014*) were significantly upregulated at the last stage of SYN induction (*Figure 1G*). IF staining confirmed the presence of SCX$^+$COL1$^+$ and TNMD$^+$COL1$^+$MKX$^+$ cells at the end of the induction (*Figure 2I*). However, expression of tenogenic markers was heterogeneous in different cells (*Figure 2I*). Additionally, we noticed gradual attrition of cells after SCL stage, resulting in considerably lower cell numbers by the end of the differentiation.

### ScRNA-seq reveals cellular heterogeneity at the end of induction of iPSC to syndetome-like cells

Even though gene expression and IF showed expression of early tendon markers by the end of differentiation using two iPSC lines figure supplement, cell heterogeneity was observed with IF (*Figure 2*).

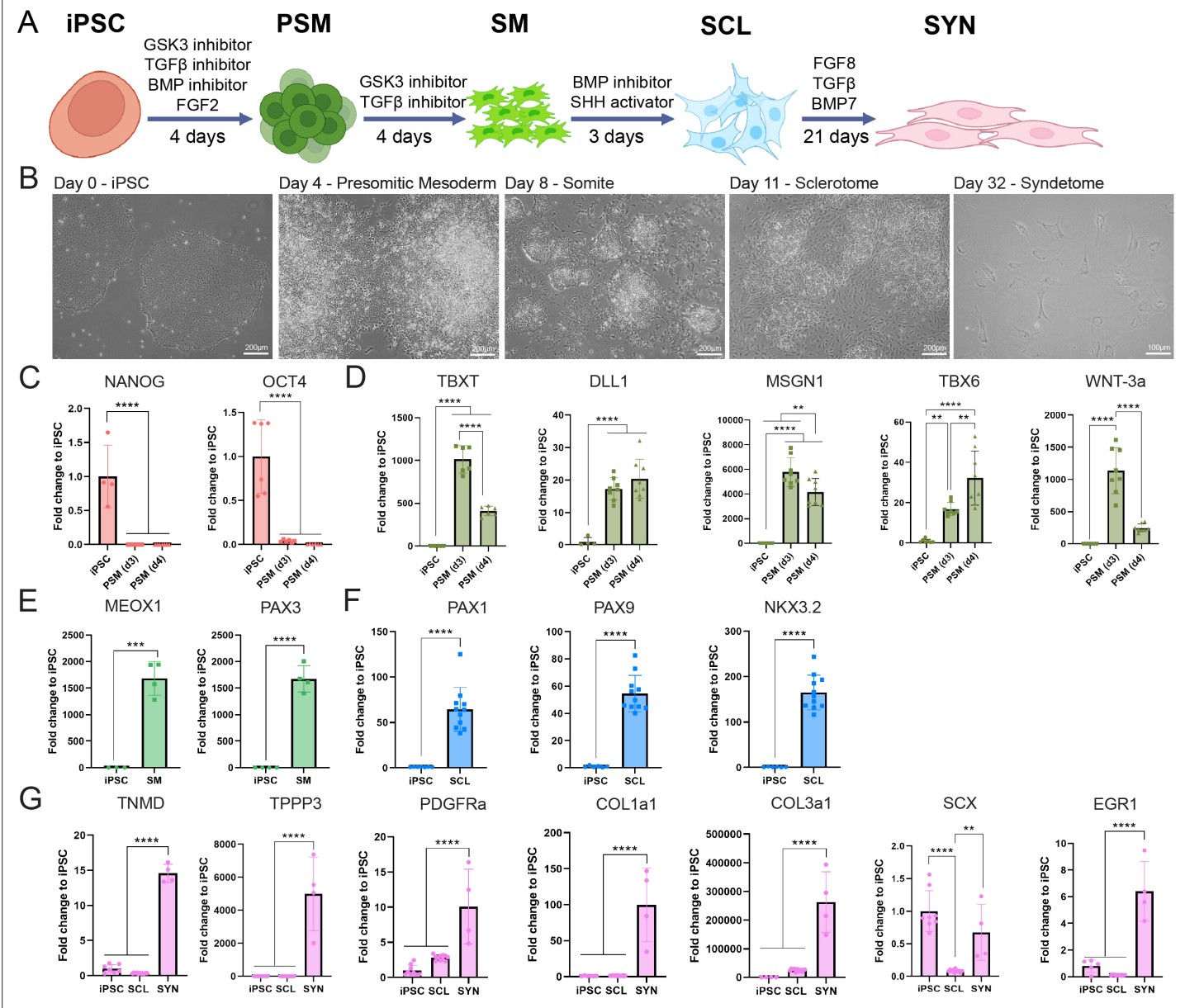

**Figure 1.** Stepwise induction of iPSCs to syndetome-like cells using chemically defined media and small molecules in vitro. (**A**) Schematic of iPSC to SYN stepwise induction using chemically defined media and small molecules. (**B**) Brightfield micrographs of cells going through the differentiation stages at. Scale bars represent 200 μm. (**C**) Pluripotent markers were expressed in iPSCs and were downregulated in further stages. Gene expression analyses for stage-specific markers with the 007i iPSC line: upregulation of early mesoderm markers at the presomitic mesoderm (PSM, $n = 8$ replicates/group) (**D**), somitogenesis at SM ($n = 8$ replicates/group) (**E**), and sclerotome-related markers at SCL ($n = 12$) (**F**). (**G**) Tenogenic markers are significantly upregulated at the SYN stage ($n = 4$) compared to iPSC ($n = 9$) and SCL ($n = 12$) stages. Differentiation experiments were repeated independently $n = 2$ with the 007i line and $n = 2$ with a second iPSC line that was later tested (83i); *$p < 0.05$, **$p < 0.01$, ***$p < 0.001$, ****$p < 0.0001$.

The online version of this article includes the following figure supplement(s) for figure 1:

**Figure supplement 1.** Flow cytometry of DLL-1 at the presomitic mesoderm (PSM) stage for low and high seeding densities.

Further, SYN differentiation efficiency was suboptimal, resulting in considerable cell attrition by the end of the induction process (*Figures 1B and 2G–I*). To further explore the observed cellular heterogeneity, we sought to investigate the cell transcriptome at each stage (iPSC, PSM, SM, SCL, and SYN) at a single-cell level using the GMP-ready line 007i and scRNA-seq analysis. The dimensional reduction based on the unsupervised Uniform Manifold Approximation and Projection (UMAP) method sorted cells into 11 clusters that could be classified as 6 cell subpopulations (*Figure 3A*). Raw counts

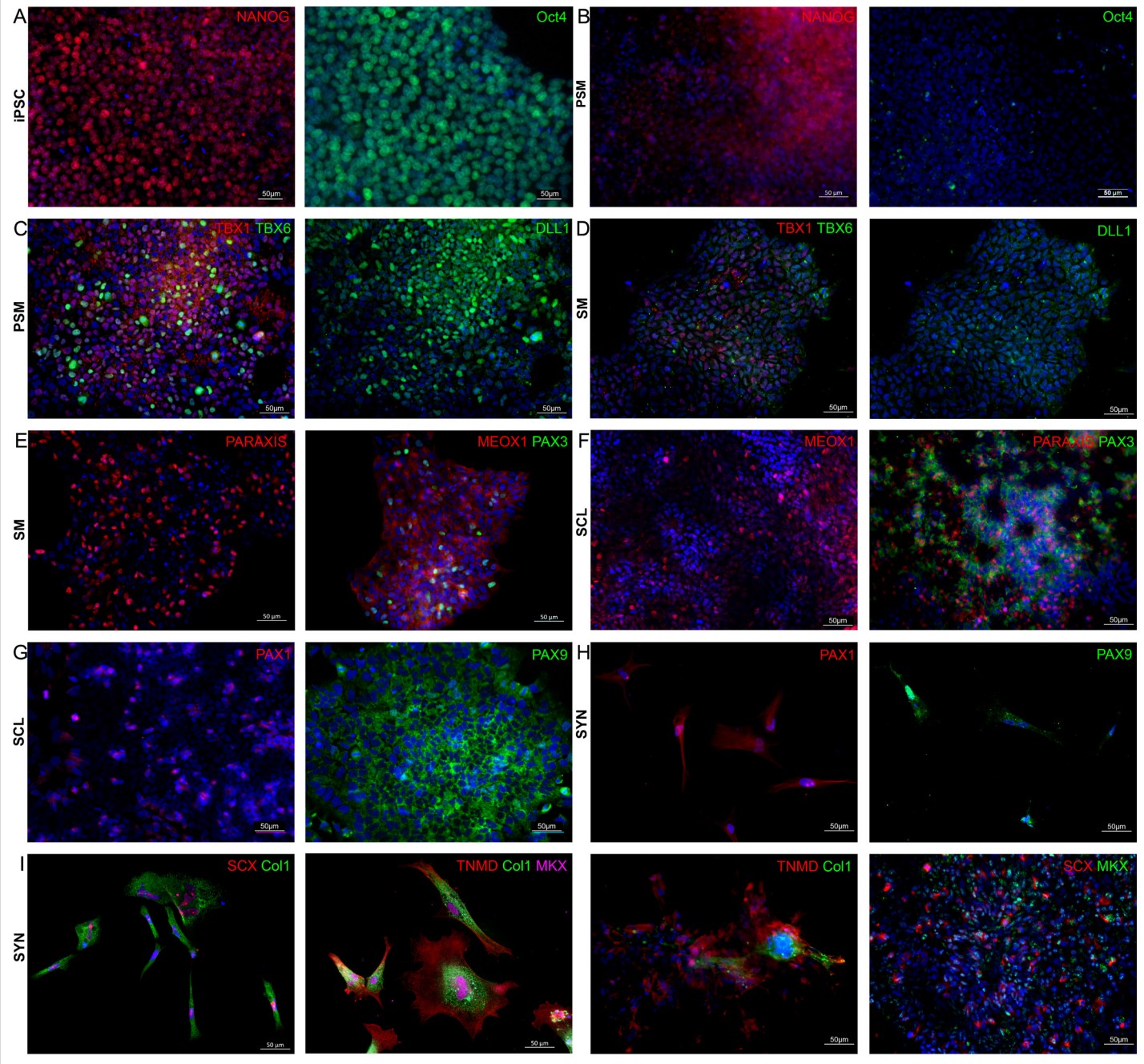

**Figure 2.** Immunofluorescence co-staining for stage-specific markers for confirmation of protein expression at each induction stage. (**A, B**) OCT4 and NANOG were observed in iPSCs but not at PSM. (**C, D**) Early mesoderm markers TBX1, TBX6, and DLL1 at PSM vs. SM. (**E, F**) Somitogenesis markers PARAXIS, MEOX1, and PAX3 at SM vs. SCL. (**G, H**) sclerotome-related markers PAX1 and PAX9 at SCL vs. SYN. (**I**) Tenogenic markers SCX, COL1, TNMD, MKX co-expression at the end of induction to SYN. Nuclei were stained with DAPI (blue). Scale bars represent 50 µm. Differentiation experiments were repeated independently $n = 2$ with the 007i line and $n = 2$ with a second iPSC line that was later tested (83i). IF staining was performed in $n = 3$ technical triplicates.

The online version of this article includes the following figure supplement(s) for figure 2:

**Figure supplement 1.** Immunofluorescence staining of selected markers and DAPI for nuclear staining from **Figure 2** expanded to show all separate channels.

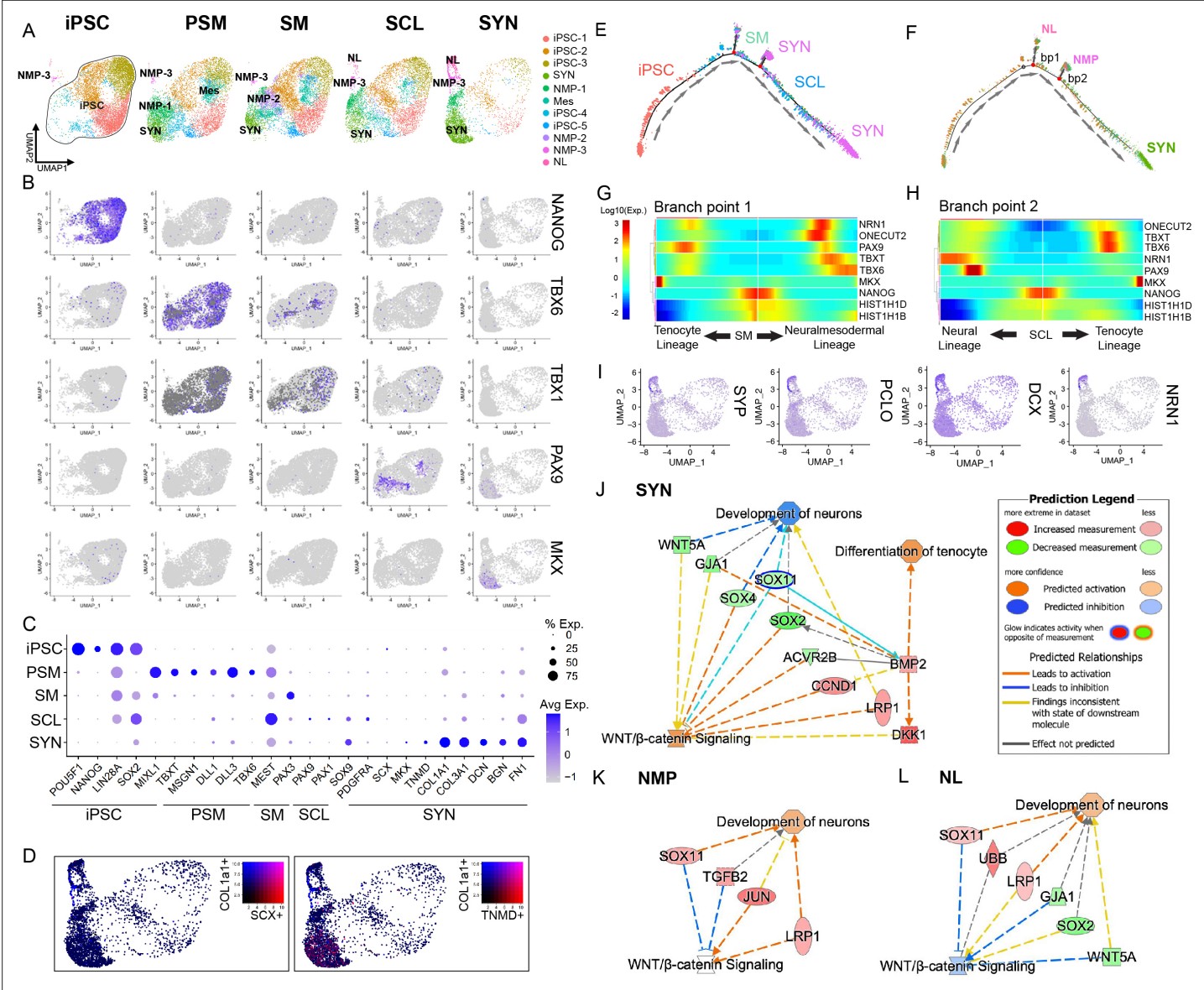

**Figure 3.** Single-cell RNA sequencing reveals cellular heterogeneity at the end of induction of iPSC to syndetome-like cells and off-target differentiation to neural-like progenitors. (**A**) UMAPs of each differentiation step sorted into 11 cell clusters from iPSC to SYN were annotated to 6 distinct cell populations: iPSC (OCT4+SOX2+NANOG+), Syndetome (SYN, MKX+TNMD+COL1A1+), Neuromesodermal Progenitors/Neural Crest cells (NMP/NC, PAX3+NRP2+COLEC12+), Mesoderm (Mes, DLL1+DLL3+PARAXIS+), Neuromesodermal Progenitors – Cranial (NMP-C, DLL1+DLL3+NOTCH1+CRAB1+), and Neural Lineage cells (NL, NRN1+DCX+NNAT+). During induction, pluripotent clusters gradually disappeared, and three main clusters emerged: SYN, NMP-C, and NL. Feature plots (**B**) and dot plots (**C**) of stage-specific genes displayed for all differentiation stages. (**D**) Expression of primary tenogenic markers COL1A1 (blue) and either SCX (red) or TNMD (red) on UMAP plot. (**E, F**) Original samples and clusters were ordered on pseudo-time developmental trajectory. (**E**) Trajectory analysis based on original samples showed transition from iPSC to SYN correlated with the samples; however, SYN cells were located within three endpoints. (**F**) Trajectory analysis based on clusters showed SYN cluster as the main differentiation endpoint with NMP-C and NL as off-target differentiation endpoints. Branching point heat maps (**G, H**) and gene expression were predominated by neural-related markers SYP, PCLO, DCX, and NRN1 (**I**). (**J, K**) IPA analysis revealed that the off-target clusters NMP-C and NL clusters were linked with increased Wnt pathway activity (**K, L**), while the SYN cluster was associated with tenocyte differentiation and linked to decreased Wnt pathway activity (**J**).

The online version of this article includes the following figure supplement(s) for figure 3:

**Figure supplement 1.** Distribution of cell subpopulations per sample.

**Figure supplement 2.** Addition of WNTi to the differentiation resulted in less off-targets.

**Figure supplement 3.** A non-biased gene ontology (GO) analysis was performed.

**Figure supplement 4.** Feature plots of NKX3.2 and MEOX1 displayed for all differentiation stages.

of cell numbers for each cluster and stage are shown in *Supplementary file 1* and normalized proportions per cluster in *Figure 3—figure supplement 1*. Feature plots and dot plots showed upregulation of stage-specific genes (*Figure 3B, C*). That is, pluripotency markers (OCT4, NANOG, SOX2) (*Takahashi and Yamanaka, 2006*) were expressed at iPSC and disappeared at the PSM stage, while PSM markers TBXT, TBX6, MSGN1, DLL1, and DLL3 (*Tani et al., 2020*; *Loh et al., 2016*; *Yamaguchi et al., 1999*) were upregulated. SM markers (MEOX1, PAX3) (*Tani et al., 2020*; *Loh et al., 2016*) and SCL markers (PAX1, PAX9, NKX3.2, SOX9) (*Tani et al., 2020*; *Loh et al., 2016*) were upregulated in a stepwise manner. Tenogenic markers (SCX, MKX, TNMD, COL1A1, DCN, COL3A1) (*Liu et al., 2014*; *Dunkman et al., 2014*) were found in SYN cluster C3 (*Figure 3B–D*). These plots show that the SYN cluster accounted for 48.4% of the entire population (*Figure 3—figure supplement 1*). Notably, two 'side-arm' clusters, which became prominent in SCL, comprised 18.3% of the cells collected at the SYN stage (*Figure 3A*, *Figure 3—figure supplement 1*). These were denoted as neuromesodermal progenitor – cranial cell populations hereafter referred to as NMP-C (DLL1$^+$DLL3$^+$NOTCH1$^+$CRABP1$^+$), (*Loh et al., 2016*; *Hofmann et al., 2004*; *Ishii et al., 2012*) and neural lineage, hereafter referred to as NL (NRN1$^+$DCX$^+$NNAT$^+$) (*Lin et al., 2010*; *Zhao and van Praag, 2020*; *Yao et al., 2018*) cell populations.

### Trajectory analysis shows off-target differentiation to neural-like progenitors

Monocle package (v2.22.0) was used to create pseudo-time trajectory (*Figure 3E, F*) and branching point heatmaps (*Figure 3G, H*). Examination of branch point heatmaps and marker expression showed that the 'side-arm' clusters that predominated in the later SCL stage and especially in SYN (*Figure 3A*) were enriched for neural-related markers NRN1, SYP, PCLO, and DCX (*Zhao and van Praag, 2020*; *Yao et al., 2018*; *Berg et al., 2019*; *Figure 3I*). This suggests that these two clusters were likely side products that deviated from the main differentiation trajectory into the neural lineage cell population.

To understand the pathways enriched at each induction stage, we used QIAGEN Ingenuity Pathway Analysis (*Figure 3J–L*). Examining the canonical pathways in our dataset, we found that in the SYN cluster, BMP activation was linked to differentiation toward the tenocyte lineage and concurrent WNT inhibition through DKK1 upregulation (*Figure 3J*). Moreover, the side arm clusters showed activation of WNT signaling and upregulation of gene signatures associated with neuronal development (*Figure 3K, L*). In conclusion, iPSCs were successfully differentiated into SYN; however, cells expressing neural markers also appeared as side products during the differentiation, and WNT signaling was found to play a role in this process.

### Inhibition of WNT signaling resulted in a decrease in off-target differentiation

Pathway analysis showed that the crosstalk between BMP and WNT may play an important role in off-target differentiation products. Thus, we hypothesized that inhibition of WNT signaling may improve induction toward tenogenesis. We investigated the addition of a potent PORCN inhibitor, WNT-C59, which blocks both canonical and non-canonical WNT signaling (Figure 5A). We observed that when the WNT inhibitor (WNTi) was implemented after day 8 (SM stage), it resulted in decreased expression of neural markers NRN1, DYNLL1, and FMN1 (*Figure 4A*). Immunostaining for neural markers (DCX, SYP, and NRN1) showed decreased presence in WNTi-treated cells (*Figure 4C1–C3*) unlike non-treated cells (*Figure 4B1–B3*). Implementation of the WNTi inhibitor also resulted in significantly higher expression of tenogenic marker TNMD partway through SYN induction (*Figure 4A2*). Notably, in some cases we observed overgrowth of cells into spherical 3D structures (*Figure 4D*), which could be related to the other cell populations comprising ~50% of this group (*Supplementary file 1* and *Figure 3—figure supplement 1*).

To further unveil the role of continuous WNT inhibition at the SM stage and until the end of the induction to SYN, we performed scRNA-seq (*Figure 5A*). 5645 cells were subjected to unsupervised clustering and plotted with UMAPs. Integrated UMAP plots of all the cell populations in this study revealed 12 distinct clusters which were further characterized. We annotated seven cell populations by using classical developmental markers and performing gene ontology enrichment analysis for differentially expressed genes (DEGs, *Figure 5B*). Six clusters expressed pluripotency and stem cell markers, including NANOG, OCT4, and SOX2, and were annotated as pluripotent cells (iPSC1-6).

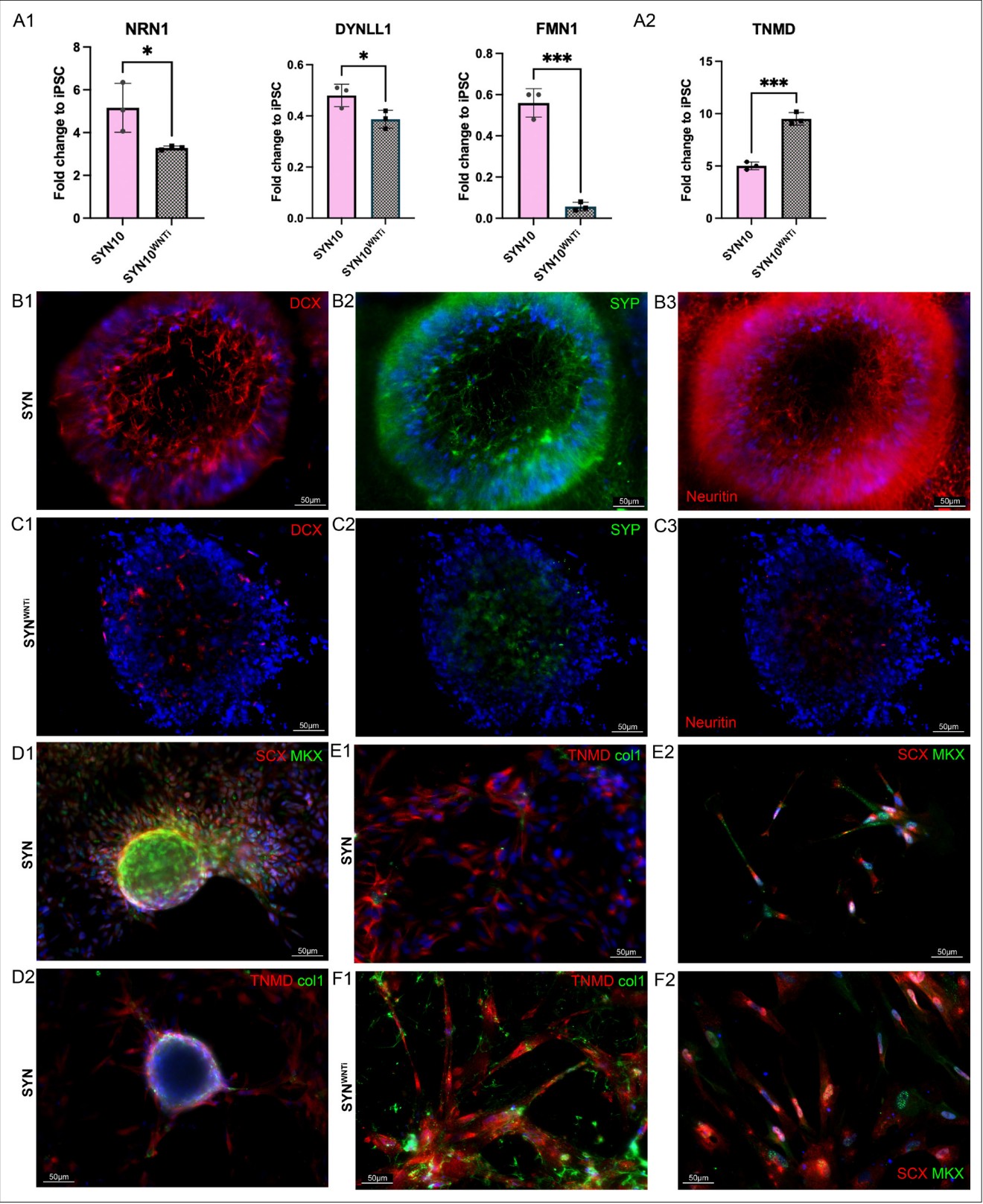

**Figure 4.** Inhibition of WNT signaling resulted in decreased expression of off-target markers and a more homogeneous population. (**A**) Gene expression analysis of neural markers NRN1, DYNLL1, and FMN1, as well as tenogenic marker TNMD on day 10 of SYN induction (day 21 of the differentiation). (**A1**) Cells treated with WNTi had significantly downregulated neural marker expression compared to just SYN. $n = 4$ replicates/group. (**A2**) Tendon gene expression was significantly upregulated in the WNTi-treated group; $n = 3$/group; *$p < 0.05$, **$p < 0.01$, ***$p < 0.001$, ****$p < 0.0001$.

*Figure 4 continued on next page*

*Figure 4 continued*

(**B, C**) Immunofluorescence staining for neural markers (DCX, SYP, NRN1), SYN (**B1–B3**), and SYN[WNTi] (**C1–C3**) showed that they were present in SYN and almost disappeared in SYN[WNTi]. Scale bars represent 50 μm. (**D–F**) Immunofluorescence staining for tenogenic markers (SCX, MKX, TNMD, COL1) of SYN vs. SYN[WNTi] groups. Scale bars are 50 μm. Differentiation experiments were repeated independently *n* = 2 with the 007i line and *n* = 2 with a second iPSC line that was later tested (83i). IF staining was performed in *n* = 3 technical triplicates.

These populations were evident during the first stages of induction, and they gradually receded. At PSM stage, two mesodermal populations were identified; one was annotated as Mes (TBXT⁺DLL1⁺) and another as NMP-C, where TWIST1 (*Loh et al., 2016*), SP5 (*Park et al., 2013*), and SNAI2 (*Shi et al., 2011*) were upregulated. These two populations further expanded at the SM stage. At the SCL stage, two additional newly formed clusters appeared. These were categorized as NL and neural crest (NC) cells. The NL cells expressed markers associated with neurogenesis, including CRABP1, DCX, and MAP2 (*Dehmelt and Halpain, 2005*). The NC cells expressed SOX4, SOX11, and NCAM1 (*Simões-Costa and Bronner, 2015*; *Deak et al., 2005*). At the SYN stage, we identified a SYN population with cells expressing tendon and ECM associated markers (COL1A1⁺, COL1A2⁺, COL3A1⁺, COL11A1⁺, FN1⁺, FBN1⁺, DCN⁺). WNT inhibition resulted in the elimination of the NC population, while the SYN population has increased in size, and a second population expressing tendon and some chondrogenic markers was evident, denoted as fibrocartilage (FC) cluster. The expression levels of classical stage-specific markers in SYN[WNTi] presented in a dot plot showed increased expression compared to untreated SYN group (*Figure 5C*). Density plots of cells expressing COL1A1 and tendon markers SCX and TNMD showed increased positive cells for those markers in SYN[WNTi] compared to SYN group (*Figure 5D*). Trajectory analysis for the WNTi-treated group showed elimination of the bifurcations that were observed in the non-treated group (*Figure 5E*). Cell proportions for each cluster were calculated by normalizing to total cell numbers. It was shown that 75.6% of all cells in SYN[WNTi] were clustered at the two SYN subpopulations, SYN and FC (66.9 and 8.7%, respectively; *Figure 5F*). Comparison of WNTi treated groups using a second iPSC line (83i) showed that in the second line, 78% of cells were clustered in the SYN group, while the FC cluster was non-existent (*Figure 3—figure supplement 2* and *Supplementary file 1*).

IPA network analysis following treatment with the WNTi showed that WNT pathway activation is associated with the NMP-C and NC clusters and not with the SYN, NL, and FC clusters (*Figure 5G*).

## Discussion

In this study, informed by ontogenesis and recent reports modeling PSC differentiation in vitro, we successfully differentiated human iPSCs to syndetome-like cells in a stepwise manner using a GMP-ready line and chemically defined serum-free media through balancing the BMP, TGFβ, Activin/Nodal, FGF, and WNT signaling pathways. iPSCs were differentiated to PSM, somite, sclerotome, and syndetome-like cells (*Figure 1A, B*). Gene expression analysis, IF, and scRNA-seq analysis validated the stepwise differentiation using classical markers that define each developmental step. However, gene regulatory network analysis revealed that the major off-target cell populations were highly associated with WNT signaling activation beginning from the SCL stage and into the SYN stage. Using scRNA-seq, we confirmed our hypothesis that WNT inhibition with the WNT signaling inhibitor (WNTi) Wnt-C59 blocked cell fate bifurcation to the 'unwanted' neural fates and drove induction toward tenogenesis.

Gene expression analysis showed significant upregulation of stage-specific markers (*Figure 1C–G*). PSM induction was assessed via the Notch receptor ligand Delta-like protein-1 (DLL1) gene and protein expression. Somites form from the epithelization and segmentation of paraxial mesoderm along the rostro-caudal (RC) axis of the embryo in the process of mesenchymal-to-epithelial transition (MET). Studies in several animal models have shown that somite RC polarity is established prior to SM formation and it is orchestrated by NOTCH signaling (*Tani et al., 2020*). Many DLL-family proteins, which are NOTCH receptor ligands, are central to this process, with DLL1 shown as the most critical during PSM formation (*Hofmann et al., 2004*; *Teppner et al., 2007*). Here, we found that the initial seeding density of iPSCs impacted DLL1 expression and thus induction efficiency. Higher starting densities resulted in decreased DLL1⁺ cell ratios assessed by flow cytometry (*Figure 1—figure supplement 1*). This is in line with several reports that have identified starting PSC seeding density as an additional

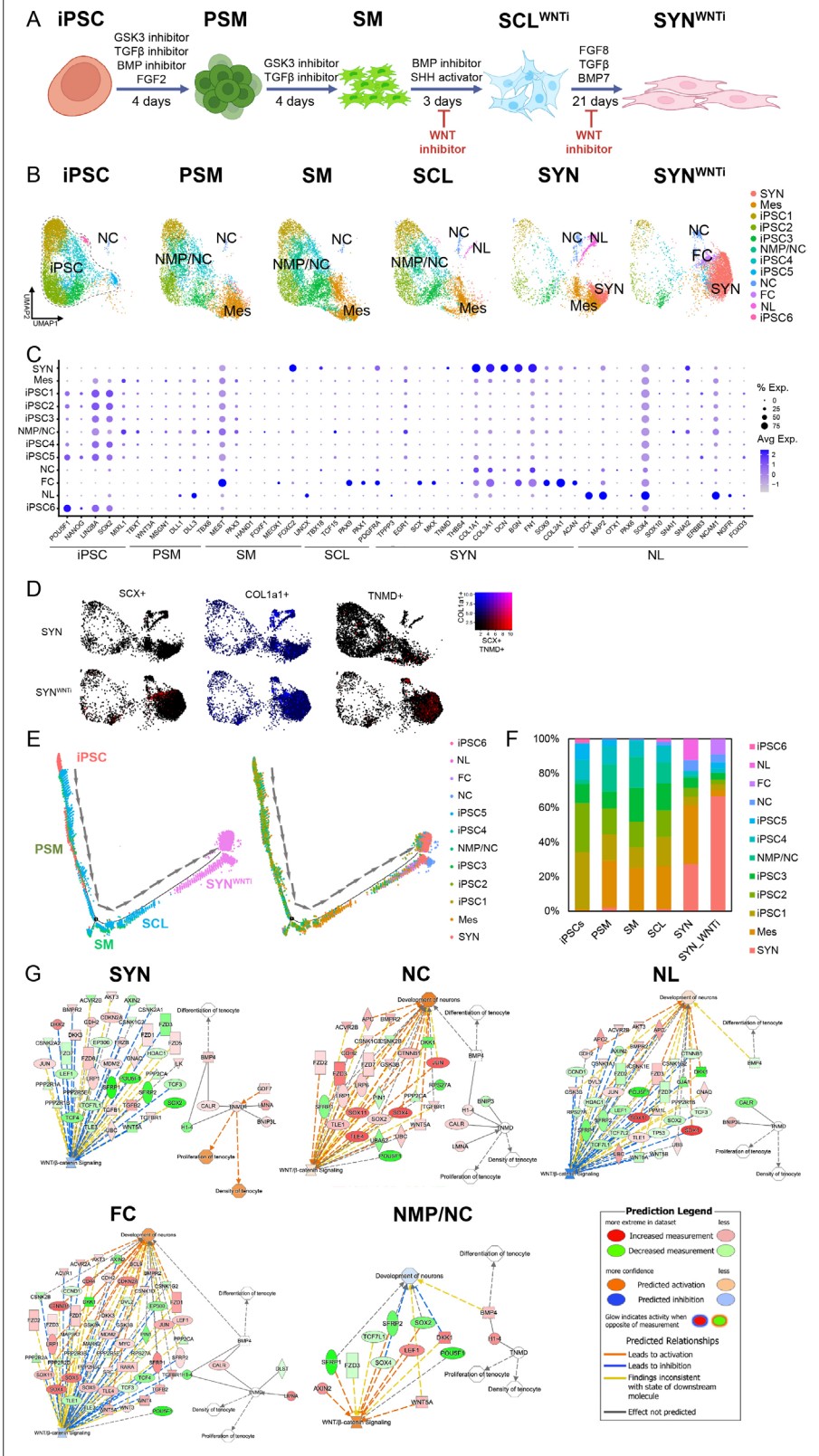

**Figure 5.** Addition of WNTi to the SCL and SYN induction stages of the differentiation improved differentiation to SYN and eliminated off-target clusters. (**A**) Schematic of optimized iPSC to SYN stepwise induction with WNTi addition implemented at SM to SCL and SCL to SYN stages. Informed by single cell transcriptomics, the addition of Wnt pathway inhibitor at the later stages of the differentiation resulted in

*Figure 5 continued on next page*

*Figure 5 continued*

a more specific differentiation of iPSCs to tenocytes. (**B**) UMAPs of each differentiation step sorted into 12 distinct clusters from iPSC to SYN were annotated into 6 distinct cell populations: Syndetome (SYN, MKX⁺TNMD⁺DCN⁺BGN⁺), iPSC (OCT4⁺NANOG⁺LIN28A⁺SOX2⁺), Neuromesodermal Progenitors/Neural Crest (NMP/NC, TBXT⁺TWIST1⁺SP5⁺SNAI2⁺), Neural Crest (NC, PTN⁺NTKR2⁺SOX4⁺SOX11⁺), Fibrocartilage (FC, COL2A1⁺SOX9⁺FN1⁺BGN⁺COL1a1⁺), and Neural Lineage (NL, SOX2⁺DCX⁺MAP2⁺UNCX⁺SOX4⁺). UMAP comparison of the SYN and SYN^WNTi populations demonstrated increased size of the SYN cluster and elimination of NL cluster. (**C**) Dot plots of gene expression of stage-specific genes for SYN and SYN^WNTi. (**D**) Expression of primary tenogenic markers Col1a1 (blue), Scx (red), and Tnmd (red) on UMAP plot for SYN and SYN^WNTi. (**E**) Original samples and clusters were ordered on pseudo-time developmental trajectory. Trajectory analysis revealed one main endpoint, the SYN cell populations. (**F**) After addition of WNTi, the proportion of cells in the SYN clusters increased by 59% while the proportion of cells in the NL cluster was eliminated. (**G**) IPA network analysis showed that WNT pathway was enriched in the NMP/NC and NC clusters but not in NL.

The online version of this article includes the following figure supplement(s) for figure 5:

**Figure supplement 1.** Dot plot of stage-specific markers of SYN^WNTi for each cluster.

**Figure supplement 2.** Change in SCX expression throughout SYN induction with WNTi.

**Figure supplement 3.** Separate and combined feature plots of SCX and TNMD expression at SCL through SYN stage with and without WNT inhibitor (WNTi).

factor influencing cell differentiation in animal models and human cells (*LeBlanc et al., 2022*). This could be attributed to auto/paracrine factors and cell-cell contact (*Wu et al., 2015*). Unfortunately, iPSC seeding density is often not reported in published protocols and it may be contributing to discrepancies and reproducibility concerns between different laboratories. Taken together, our data indicate that starting seeding density is a crucial component of chemical induction which could influence reproducibility and translatability of a given approach.

Somitogenesis induction was achieved by TGFβ inhibition (SB431542) and moderate WNT activation (CHIR99021) similar to previous reports (*Nakajima et al., 2018*). Combined SHH signaling activation (SAG) and BMP inhibition (LDN193189) resulted in induction of SM to SCL. Last, stepwise FGF8 followed by BMP and TGFβ administration induced SYN formation. Early tendon-associated markers were significantly upregulated at the final timepoint of SYN induction (*Figure 1G*). FGF ligands including FGF8 are secreted by the myotome region of the somite, located next to the sclerotome, inducing scleraxis (SCX) expression, which in turn drives syndetome specification (*Tani et al., 2020*). Protein expression at each stage was verified with IF (*Figure 2*). However, at the end of differentiation we noticed gradual cell attrition in the cultures (*Figures 1B and 2I*). This could be attributed to the heterogeneity of the cell population at SCL stage, which was confirmed by scRNA-seq (*Figure 3A*).

In the current study, scRNA-seq analysis showed that mesodermal cells persisted until SYN induction (*Figure 3A–C*). DLL1 expression was significantly upregulated at PSM and downregulated at SM stage. At PSM, 96.2% of the cell population was DLL1⁺ (*Figure 1—figure supplement 1*). However, dot plots showed some DLL1 and DLL3 expression in all stages and clusters (*Figure 3C*, *Figure 3—figure supplement 1*). Consequently, using a single marker may not be sufficient to assess lineage differentiation efficiency. Trajectory analysis showed two distinctive bifurcations at SM and SCL stages. The first branchpoint was observed at the SM stage, when a Mes (DLL1⁺DLL3⁺PARAXIS⁺) and a transient NMP (PAX3⁺NRP2⁺COLEC12⁺) cell population were identified. NMPs were present at SM and disappeared thereafter, while at SCL two 'side-arm' clusters evolved (*Figure 3A*). The NL cluster was enriched for several neural-associated markers (NRN1⁺NNAT⁺CRABP1⁺ONECUT2⁺NPTX2⁺). In the NMP-C side-arm cluster the top DEGs were a combination of neural-associated markers (NOTCH1⁺CRABP1⁺GREB1⁺ID4⁺), cranial mesodermal markers (DLL1⁺DLL3⁺), and ECM genes including collagens and FN1. The emergence of these two off-target cell populations became quite prominent at SCL, while they were further amplified at SYN (*Figure 3A*).

Sclerotome is derived from the ventromedial part of somites and its induction is orchestrated by SHH (*Tani et al., 2020*). It is established that SHH and NOG produced in the notochord and neural tube stimulate MET, which produces both NC cells from the ectoderm as well as SCL at the ventromedial aspect of somites (*Kahane and Kalcheim, 2020*). Different groups have achieved sclerotome induction of iPSCs with varying approaches. For instance, *Loh et al., 2016* identified that WNT and SHH acted antagonistically in SCL specification. The NMP-C off-target cell population that was first

observed in the SCL and SYN groups could have arisen from NMPs (*Figure 3A*). NC, which are a transient cell population of ectodermal origin, are derived from bipotential NMPs from the primitive streak in the presence of FGF, WNT, and likely medium BMP activity (*Piacentino and Bronner, 2018*; *Wymeersch et al., 2021*). However, the timing and levels of BMP signaling required for NC differentiation are still not well understood. (*Wu et al., 2021*) observed NC off-target populations during stepwise iPSC to somite to chondroprogenitor differentiation. In that study, the authors first observed NC at the chondroprogenitor stage, and hypothesized that they emerged due to BMP treatment for 3 days at the SCL stage (*Wu et al., 2021*). In the present study, we observed the emergence of NC cells earlier, and even though we applied a BMP inhibitor during the SM to SCL differentiation, it may not have been sufficient to modulate BMP activation. Lastly, the NC off-target population could have been derived from NMP/NC cells or from NMPs present at SM. In vivo somites give rise to the dermatome and myotome dorsally and the SCL ventromedially (*Tani et al., 2020*). The dermatome sits next to the ectoderm and can differentiate into the meningotome, which can form spinal meninges as well as fibroblasts of the meninges and arachnoids (*Christ et al., 2004*). The off-target neural cluster in the present study showed increased expression of markers associated with NC as well as neural markers. Thus, it could be posited that it was derived from the NC off-target subpopulation identified at SCL.

Next, DEG analysis using the IPA platform revealed the interaction between WNT and BMP signaling at the cornerstone between the SYN cluster and the NL and NC off-target 'side-arm' clusters (*Figure 3J–L*). More specifically, upregulation of genes associated with WNT signaling activation in both off-target clusters was observed. Therefore, a WNT pathway inhibitor was employed during SCL and SYN differentiation, which resulted in significant downregulation of selected neural markers, assessed by RT-qPCR and IF (*Figure 4*).

Treatment with the WNTi following the SM stage resulted in the expansion of the SYN clusters (*Figure 5A*) as well as complete elimination of the NL off-target cluster (*Figure 5A*). Further, a new cell population was observed only in the WNTi-treated group, expressing some chondrogenic markers (COL2A1$^+$SOX9$^+$FN1$^+$BGN$^+$COL1A1$^+$), tenogenic genes, and Klf transcription factors, which were recently shown to be associated with bipotential tendon-to-bone attachment cells (*Kult et al., 2021*). In the same study, these cells were shown to activate a combination of tendon and chondrogenic transcriptomes. Fibrocartilage is found in the enthesis of tendons and shares the same progenitors as the syndetome (*Blitz et al., 2013*; *Sugimoto et al., 2013*). Intriguingly, the FC cluster was not observed in the second cell line we differentiated (*Figure 3—figure supplement 2*), and it was the only observed difference between the two lines. This could have also been related to lineage bias, since the 007i line had been derived from PBMCs while the 83i line has been derived from fibroblasts (*Kim et al., 2024*). Fibrocartilage progenitors express SCX; however, divergence of SCX$^+$ cells toward tendon or fibrocartilage has yet to be fully understood. In this work, WNT inhibition at the SM stage resulted in increased numbers of SCX$^+$ derivatives compared to SYN while eliminating the NL population. However, the NC cell population was not diminished, indicating that these cells might have been generated due to 'contaminating' multipotential NMPs still present at SM and SCL. IPA network analysis revealed WNT pathway activation in mesodermal and NC clusters and not the final products of the induction, that is, NL, FC, and SYN clusters (*Figure 5G*). This further confirms our hypothesis regarding its activation following SM specification and suggests that it plays an important role in specifying the fate of NMPs toward neural lineages. Further studies are needed to delineate the level of BMP activity required to either induce or block bifurcation of NMPs to NC fates. Interestingly, SCX expression peaked in the middle of SYN induction and decreased at the end (*Figure 5—figure supplement 2*), when TNMD, COLA1A, and other tenogenic markers (collagens, DCN, BGN1, FN1) were upregulated. This suggests that SYN induction could be further optimized and shortened. Further, WNT inhibition during the entirety of the SYN induction might potentially negatively impact SYN maturation, and it should be further fine-tuned. For instance, WNT signaling has been shown to upregulate TNMD expression in equine BM-MSCs (*Miyabara et al., 2014*), while it was shown to suppress TNMD and other tenogenic genes in tendon-derived cells in vitro (*Kishimoto et al., 2017*).

Previous studies investigating iPSC to SYN differentiation reported varying yields at the end of induction. *Nakajima et al., 2018* reported 68% SCX + cells using flow cytometry. In a follow-up study, the same research group reported 91.6% SCX+, 90.4% MKX+, 79.9% COL1A1+and 77.5% COL1A2+ protein expression using IF quantification (*Nakajima and Ikeya, 2021*). However, although

informative, it could be noted that IF is only a semi-quantitative assessment burdened with operator bias and lower sensitivity compared to flow cytometry or scRNA-seq, unless performed in a more automated manner (*Lara et al., 2021*). Nevertheless, despite those caveats, the high individual expression of all four markers (77.5–91.6%) with IF supports a relatively efficient SYN differentiation overall. *Kaji et al., 2021* differentiated mESCs from SCX-GFP reporter mice to SYN and reported 90% efficiency assessed by flow cytometry for SCX, while Sugimoto et al. showed 66% efficiency utilizing a similar Scx-GFP mouse model combined with flow cytometry and scRNA-seq assessments (*Kaji et al., 2021*; *Yoshimoto et al., 2022*). In the current study, SYN clusters, characterized by scRNA-seq analyses, greatly increased in the SYN^WNTi group, from 47.6% prior to WNTi application to 67–78% of total cells after treatment (*Figure 5F*, *Figure 3—figure supplement 2*). In conclusion, we showed that WNT inhibition was able to successfully modulate the later stages of differentiation from SM to SYN.

This study is not without limitations. The IPA network analysis is a knowledge-based and hypothesis-driven platform. We have specifically targeted known pathways to be involved in syndetome differentiation. However, WNT signaling stood out with very specific affinity to the off-target populations, and we have verified our findings with experiments proving this hypothesis. In the current study, IF was used solely as a qualitative assessment of protein presence and localization to support our main quantitative findings that were used to optimize the induction (gene expression, flow cytometry, single-cell transcriptomics). However, IF could also be used quantitatively if a more unbiased quantitative approach is employed as described earlier. Further, the SCL to SYN induction, maturation, and expansion in culture was considerably long and resulted in decreased expression of SCX (*Figure 5—figure supplement 2*), suggesting that it could be shortened. The objective of this study was to test and optimize the induction protocol in two different lines: one that has been widely tested and published (83i derived from fibroblasts) and a GMP-ready line (007i derived from PBMCs). The difference between the two lines could be further explored, since it could be the result of lineage bias. Even though following WNTi treatment, induction efficiency increased considerably, there were still other populations present, including NC and undifferentiated cells. Cell sorting could be used to isolate a more homogeneous cell population. Further, removal of undifferentiated cells is crucial for in vivo preclinical studies. Future work is warranted to assess the functionality of the cells in vivo.

## Conclusions

Tendons have poor innate healing capacity, and novel approaches are urgently needed. Tendon cell therapy offers potential for regeneration of injured tissues. In this study, we successfully differentiated a GMP-ready human iPSC line to syndetome-like cells in a stepwise manner using fully defined media. However, scRNA-seq trajectory analysis revealed off-target differentiation toward a neural phenotype, resulting in heterogeneous differentiation. DEG and regulatory network analyses demonstrated WNT-associated effectors implicated in the off-target groups, which prompted us to apply a WNT inhibitor at the somite stage. Taken together, our data provide evidence that by manipulating WNT signaling, we can achieve a more specific and robust differentiation of iPSCs to tendon progenitors. Elucidating the mechanism of the WNT signaling pathway using a development-inspired protocol can lead to the development of more powerful and specific differentiation protocols for cell therapy applications. iPSC-derived tendon progenitors can be an off-the-shelf cell source for tendon and ligament injuries that are currently untreatable or have poor surgical outcomes.

## Materials and methods

### Resource availability

#### Lead contact

Further information and requests for resources, reagents, data, and code should be directed to the corresponding author, Dmitriy Sheyn (Dmitriy.Sheyn@csmc.edu).

#### Materials availability

This study did not produce any unique reagents or materials.

## Experimental model and subject details

### iPSC maintenance and expansion

Human iPSC lines were obtained from the Cedars-Sinai Core facility and were expanded on Matrigel-coated plates (BD Biosciences) (0.08 mg/well in 6-well plates). Cells were fed every other day with cGMP mTeSRPlus media (StemCell Technologies) and passaged with ReLeSR (StemCell Technologies). Two distinct fully characterized hiPSC lines were used in this study: the GMP-ready CS0007iCTR-n5 line and the CS83iCTR-22n1 line (available here).

## Method details

### iPSC to SYN differentiation

iPSCs were seeded into Matrigel-coated plates (#354230, Corning, Corning, NY) (0.08 mg/well in 6-well plates and 0.04 mg/well in 12-well plates) and were induced to PSM at 30% confluency (~50–60 aggregates/cm$^2$) in basal medium (BM) composed of IMDM/Ham's F12 (1:1) (Thermo Fisher), 1% lipid concentrate (Thermo Fisher), 0.5% antibiotic–antimycotic solution (Thermo Fisher), 15 µg/ml apo-transferrin (Sigma), 450 µM monothioglycerol (Sigma), 7 µg/ml insulin (Sigma) supplemented with 10 µM SB431542 (Sigma), 2 µM DMH1 (Santa Cruz Biotechnology), 20 ng/ml FGF2 (Biogems-PeproTech), and 10 µM CHIR99021 (Biogems-PeproTech) (PSM media). After 3 days, media were changed to fresh PSM media. Geltrex (#A1413301, Thermo Fisher) was diluted 1:100 in cold serum-free DMEM/F12 to coat wells in 12-well plates (0.5 ml/well). The coated plates were incubated for at least 1 hr at 37°C before use. At day 4, PSM cells were washed with PBS and lifted using Accutase (StemCell Technologies) for 5 min at 37°C and reseeded at 28,500 cells/cm$^2$ into the Geltrex-coated plates supplemented with SM differentiation media composed of BM supplemented with 10 µM SB431542, 5 µM CHIR99021, and 10 µM Y-27632 dihydrochloride (Biogems, CA, USA). On days 5 and 7, media were changed to fresh SM media without Y-27632 dihydrochloride. At day 8, cells were cultured in sclerotome (SCL) differentiation media containing BM supplemented with 100 nM SAG (Biogems-PeproTech) and 0.6 µM LDN193189 (Biogems-PeproTech). Media were changed to fresh SCL media at day 10. At day 11, cells were washed with PBS, lifted using Accutase (StemCell Technologies) for 5 min at 37°C and were reseeded at 60,000 cells/cm$^2$ into Geltrex-coated (Thermo Fisher) plates with BM supplemented with 20 ng/ml FGF8 (Peprotech). After 3 days, media were changed to BM supplemented with 10 ng/ml TGFβ3 (StemCell Technologies) and 10 ng/ml BMP7 (PeproTech) (SYN maturation, SYN-M media). Media were changed to fresh SYN-M media every 3 or 4 days until day 32.

### Application of Wnt inhibitor

Following initial single cell analyses, the iPSC to SYN induction was repeated with the addition of the Wnt signaling inhibitor Wnt-C59 (Cayman Chemical, MI, USA) beginning from the SCL induction stage and onwards; the somite cells treated with SCL media and sclerotome cells treated with SYN-M media were supplemented with 1 µM Wnt-C59.

### Flow cytometry

For assessment of DLL-1 levels to determine induction efficiency, cells that were induced for 4 days were lifted using Accutase for 5 min at 37°C and washed 3x with FACS buffer containing 2% bovine serum albumin (BSA, A4503, Sigma, St. Louis, MO) and 0.1% sodium azide (S2002, Sigma) in PBS. The cells were either unstained, labeled with DLL1 anti-human antibody (APC-Vio 770, #130-106-148, Miltenyi Biotec, Bergisch Gladbach), or labeled with its associated isotype control (APC-Vio 770, #130-113-759, Miltenyi Biotec) for 15 min protected from light at 4°C. After the incubation, cells were washed again and resuspended with FACS buffer. Data were acquired on a BD LSR Fortessa analyzer (BD Biosciences, San Jose, CA) and were analyzed using FlowJo software (FlowJo LLC, Ashland, OR).

### Gene expression analysis

Differentiation to SYN was defined based on expression of developmental stage-specific markers (*Supplementary file 1*). Cells were collected from each developmental stage, and total RNA was isolated using the RNeasy plus kit (QIAGEN) and reverse transcribed with the high-capacity cDNA reverse transcription kit (Applied Biosystems). Then, cDNA was amplified by performing qPCR with

TaqMan gene expression. The threshold cycle (Ct) value of 18S rRNA was used as an internal control using the TaqMan gene expression FAM/MGB probe system (4333760F, Thermo Fisher). The Livak method was used to calculate $\Delta\Delta$Ct values, and fold change was calculated as $2^{-\Delta\Delta Ct}$, as previously described and published (*Schmittgen and Livak, 2008*).

## Immunocytochemistry for phenotype confirmation

In preparation for immunocytochemistry, iPSCs were seeded onto coverslips. The coverslips were coated with Cultrex Poly-D-Lysine (R&D Systems) for 5 min at RT, aspirated, and air-dried for at least 2 hr. They were then coated with Matrigel as per the seeding protocol described previously. At each developmental stage, cells were fixed with 4% paraformaldehyde for 30 min at RT, followed by 3x PBS washes. Briefly, after serum-free protein blocking (X0909, Dako, Agilent), cells were hybridized with various primary antibodies (Key Resource table) overnight at 4°C. The following day, these cells were incubated with fluorophore-conjugated secondary antibodies (1 hr at 37°C). Coverslips were mounted onto slides using ProLong Gold with DAPI (Molecular Probes, Life Technologies). A Carl Zeiss fluorescence microscope (Imager, Z1, ApoTome and MBF equipped) was used to acquire images.

## ScRNA-seq sample preparation, sequencing, and data processing

At each developmental stage (iPSC, PSM, SM, SCL, SYN, SYN^WNTi), cells were washed with PBS and lifted using Accutase (StemCell Technologies) for 5 min at 37°C, and then centrifuged for 3 min at 1000 RPM. The cells were then resuspended in media, filtered using a 40 μM Flowmi cell strainer (Thermo Fisher), and washed twice more with media before being filtered for the last time to ensure a single-cell suspension. The filtered cells were manually counted in quadruplicate with 0.4% trypan blue dye (Thermo Fisher), and cells were resuspended in media at a concentration of 1500 cells/μl.

Single-cell RNA-seq libraries were prepared per the Single Cell 3′ v3.1 Reagent Kits User Guide (10x Genomics, Pleasanton, California) using the 10x Genomics Chromium Controller. Barcoded sequencing libraries were quantified by quantitative PCR using the Collibri Library Quantification Kit (Thermo Fisher Scientific, Waltham, MA). Libraries were sequenced on a NovaSeq 6000 (Illumina, San Diego, CA) as per the Single Cell 3′ v3.1 Reagent Kits User Guide, with a sequencing depth of ~40,000 reads/cell.

Raw sequencing data were demultiplexed and converted to FASTQ format by using bcl2fastq v2.20 (Illumina, San Diego, California). More than 200 million reads were obtained for each sample. Reads were mapped to the human GRCh38 genome, and count quantification was done using 10x Genomics Cell Ranger v.7.0.0.

We utilized our previously established single-cell analysis platform based on R (v4.1.2) (*Jiang et al., 2022*). The Seurat package (v4.1.0) was used to load and process the single-cell data. Specifically, the data for each sample were loaded using *Read10X*. Quality control was performed for each sample. The parameters used for quality control were *nFeature_RNA*, that is the counts of genes detected in each cell, and *percent.mt*, that is the percent of mitochondrial genes expressed in each cell. Any cells having *nFeature_RNA* outside of the range of 200–8000 were excluded in the downstream analysis. Any cells with the *percent.mt* being smaller than 20% were excluded in the downstream analysis. The Seurat object for each sample after the quality control was combined into a larger Seurat object. The large Seurat object was then normalized, anchored, and integrated. Principal component analysis with PC = 30 was performed, dimensional reduction with UMAP was applied, and clusters were identified in an unsupervised manner with a resolution of 0.5. Following the identification of clusters, each cell was annotated with two important identities: the sample they were derived from and the cluster (cell type) they belonged to. The DEGs were calculated by comparing across sample identity or cluster identity, that is, a specific cluster vs. all cells, a specific sample vs. all other samples, a specific sample vs. another specific sample, or as otherwise noted in the manuscript. The DEGs were inputted to QIAGEN Ingenuity Pathway Analysis (IPA) for gene ontology term enrichment and pathway analysis. Genes with p < 0.0001 were included in the QIAGEN IPA analysis.

The pseudo-time trajectory was constructed using the Monocle package (v2.18.0). The previously combined large Seurat object was loaded into the Monocle package. The DDRTree method was used to reduce the dimension of the data. The cells were projected on the trajectory and visualized based on their identity or marker expression.

## Statistical analyses

Data are presented as mean ± standard deviation from the mean. Normally distributed data were analyzed with an unpaired *t*-test (for two groups) or non-repeated measures analysis of variance followed by Tukey–Kramer HSD post hoc analysis when more than two groups were compared. Non-parametric data were analyzed using the Mann–Whitney and Kruskal–Wallis tests. Statistical significance was set at $p < 0.05$.

## Acknowledgements

This study was supported by Cedars-Sinai Board of Governors Regenerative Medicine Institute, the NIH/NIAMS K01AR071512 (DS), California Institute for Regenerative Medicine DISC0-14350 and EDUC4-12751 (WJ) awards. We also wish to acknowledge Cedars-Sinai Biobank and Research Pathology Resource for the help with imaging and the Cedars-Sinai Applied Genomics, Computation, and Translational Core for the help with transcriptomic analysis.

## Additional information

### Funding

| Funder | Grant reference number | Author |
|---|---|---|
| California Institute for Regenerative Medicine | DISC0-14350 | Dmitriy Sheyn |
| Cedars-Sinai Board of Governors Regenerative Medicine Institute | NIH/NIAMS K01AR071512 | Dmitriy Sheyn |
| California Institute for Regenerative Medicine | EDUC4-12751 | Wensen Jiang |

The funders had no role in study design, data collection, and interpretation, or the decision to submit the work for publication.

### Author contributions

Angela Papalamprou, Conceptualization, Data curation, Formal analysis, Supervision, Validation, Investigation, Visualization, Methodology, Writing – original draft, Project administration, Writing – review and editing; Victoria Yu, Data curation, Formal analysis, Visualization, Methodology, Writing – original draft; Wensen Jiang, Data curation, Formal analysis, Visualization, Writing – original draft; Julia Sheyn, Data curation, Formal analysis, Visualization; Tina Stefanovic, Chloe Castaneda, Melissa Chavez, Data curation, Formal analysis; Angel Chen, Data curation; Dmitriy Sheyn, Conceptualization, Resources, Data curation, Formal analysis, Supervision, Funding acquisition, Investigation, Writing – original draft, Project administration, Writing – review and editing

### Author ORCIDs

Angela Papalamprou (iD) https://orcid.org/0000-0002-0496-2501
Dmitriy Sheyn (iD) https://orcid.org/0000-0002-3333-1485

Reviewer #1 (Public review): https://doi.org/10.7554/eLife.89652.3.sa1
Reviewer #2 (Public review): https://doi.org/10.7554/eLife.89652.3.sa2
Reviewer #3 (Public review): https://doi.org/10.7554/eLife.89652.3.sa3
Author response https://doi.org/10.7554/eLife.89652.3.sa4

## Additional files

### Supplementary files

Supplementary file 1. Normalized cell counts (expressed as %) per cluster following WNTi treatment, shown for the two cell lines, 007i and 83i.

MDAR checklist

## Data availability

The raw sequencing data and original data are accessible in Gene Expression Omnibus. The series accession number is GSE229008. The original code is accessible at GitHub: https://github.com/jason199112345/scRNA-seq-for-the-project-of-iPSC-to-Tenocyte-differentiation (copy archived at *jason199112345, 2023*).

The following dataset was generated:

| Author(s) | Year | Dataset title | Dataset URL | Database and Identifier |
|---|---|---|---|---|
| Papalamprou A, Yu V, Jiang W, Sheyn J | 2023 | Single cell RNA-seq data for iPSC differentiation into Syndetome | https://www.ncbi.nlm.nih.gov/geo/query/acc.cgi?acc=GSE229008 | NCBI Gene Expression Omnibus, GSE229008 |

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

# Appendix 1

## Appendix 1—key resources table

| Reagent type (species) or resource | Designation | Source or reference | Identifiers | Additional information |
|---|---|---|---|---|
| Cell line (human) | 007i-cntr-n5 | Cedars-Sinai iPSC Core Facility | N/A | https://biomanufacturing.cedars-sinai.org/?filter_cell-type=ipsc&filter_primary-tissue&filter_disease&filter_sex&filter_age-at-sampling&filter_ethnicity&filter_race&filter_gene&filter_mutation&filter_project&cs_product_search |
| Cell line (human) | 83i-cntr-22n1 | Cedars-Sinai iPSC Core Facility | N/A | See above |
| Antibody | Rabbit Polyclonal Anti-TBX1 | Abcam | Ab19530 | IF (1:25-1:250) |
| Antibody | Goat Polyclonal Anti-TBX6 | R&D Systems | AF4744 | IF (1:25-1:250) |
| Antibody | Mouse Monoclonal Anti-DLL1 | Miltenyi Biotec | 130-106-148 | IF (1:25-1:250) |
| Antibody | Mouse Monoclonal Anti-DLL1 | Miltenyi Biotec | 130-106-148 | FACS (10 µl per test) |
| Antibody | Rabbit Polyclonal Anti-PARAXIS | Sigma | HPA060221 | IF (1:25-1:250) |
| Antibody | Rabbit Polyclonal Anti-MEOX1 | Sigma | HPA045214-25UL | IF (1:25-1:250) |
| Antibody | Mouse Monoclonal Anti-PAX1 | Developmental Studies Hybridoma Bank | Clone: C2 | IF (1:25-1:250) |
| Antibody | Mouse Monoclonal Anti-PAX3 | Thermo Fisher | 60217-1-IG | IF (1:25-1:250) |
| Antibody | Mouse Monoclonal Anti-PAX9 | Thermo Fisher | H00005083-M03 | IF (1:25-1:250) |
| Antibody | Rabbit Polyclonal Anti-NKX3.2 | Sigma | HPA027564 | IF (1:25-1:250) |
| Antibody | Rabbit Polyclonal Anti-SCX | Thermo Fisher | PA5-115874 | IF (1:25-1:250) |
| Antibody | Goat Polyclonal Anti-COL1 | Bio-Rad | 131001 | IF (1:25-1:250) |
| Antibody | Rabbit Polyclonal Anti-TNMD | Sigma | HPA055634 | IF (1:25-1:250) |
| Antibody | Mouse Monoclonal Anti-MKX | Thermo Fisher | MA5-26976 | IF (1:25-1:250) |
| Antibody | Monoclonal Mouse Anti-Neuritin | R&D Systems | AF283 | IF (1:25-1:250) |
| Antibody | Mouse Monoclonal Anti-SYP | Biolegend | 837104 | IF (1:25-1:250) |
| Antibody | Polyclonal Sheep Anti-DCX | R&D Systems | AF10025-100 | IF (1:25-1:250) |
| Antibody | Monoclonal Mouse Anti-NANOG | Sigma-Aldrich | AMAB91393 | IF (1:25-1:250) |
| Antibody | Monoclonal Mouse Anti-OCT4 | Cell Signalling | D705Z | IF (1:25-1:250) |
| Sequence-based reagent | Dynein light chain LC8-type 1 | Thermo Fisher | Hs04378026_m1 | qPCR primers |
| Sequence-based reagent | Neuritin 1 | Thermo Fisher | Hs00213192_m1 | qPCR primers |

*Appendix 1 Continued on next page*

*Appendix 1 Continued*

| Reagent type (species) or resource | Designation | Source or reference | Identifiers | Additional information |
|---|---|---|---|---|
| Sequence-based reagent | One cut homeobox 2 | Thermo Fisher | Hs00191477_m1 | qPCR primers |
| Sequence-based reagent | Formin 1 | Thermo Fisher | Hs05010770_m1 | qPCR primers |
| Sequence-based reagent | Nanog homeobox | Thermo Fisher | Hs02387400_g1 | qPCR primers |
| Sequence-based reagent | POU class 5 homeobox1 | Thermo Fisher | HS04260367_gH | qPCR primers |
| Sequence-based reagent | T-box transcription factor T | Thermo Fisher | HS00610080_m1 | qPCR primers |
| Sequence-based reagent | SRY-box transcription factor 2 | Thermo Fisher | HS04260357_g1 | qPCR primers |
| Sequence-based reagent | Delta like canonical notch ligand 1 | Thermo Fisher | HS01011330_m1 | qPCR primers |
| Sequence-based reagent | T-box transcription factor 6 | Thermo Fisher | Hs00365539_m1 | qPCR primers |
| Sequence-based reagent | Mesogenin 1 | Thermo Fisher | Hs03405514_s1 | qPCR primers |
| Sequence-based reagent | Wnt family member 3A | Thermo Fisher | Hs00263977_m1 | qPCR primers |
| Sequence-based reagent | Mesenchyme homeobox 1 | Thermo Fisher | Hs00244943_m1 | qPCR primers |
| Sequence-based reagent | Transcription factor 15 | Thermo Fisher | Hs00231821_m1 | qPCR primers |
| Sequence-based reagent | Paired box 3 | Thermo Fisher | Hs00240950_m1 | qPCR primers |
| Sequence-based reagent | Paired box 9 | Thermo Fisher | Hs00196354_m1 | qPCR primers |
| Sequence-based reagent | Paired box 1 | Thermo Fisher | Hs01071292_m1 | qPCR primers |
| Sequence-based reagent | NK3 homeobox 2 | Thermo Fisher | Hs00154168_m1 | qPCR primers |
| Sequence-based reagent | Scleraxis | Thermo Fisher | Hs03054634_g1 | qPCR primers |
| Sequence-based reagent | Tenomodulin | Thermo Fisher | Hs00223332_m1 | qPCR primers |
| Sequence-based reagent | Tubulin polymerization promoting protein family member 3 | Thermo Fisher | Hs03043892_m1 | qPCR primers |
| Sequence-based reagent | Platelet derived growth factor receptor alpha | Thermo Fisher | Hs00998018_m1 | qPCR primers |
| Sequence-based reagent | Early growth response 1 | Thermo Fisher | Hs00152928_m1 | qPCR primers |
| Sequence-based reagent | Collagen type III alpha 1 chain | Thermo Fisher | HS00943809_m1 | qPCR primers |
| Sequence-based reagent | Collagen type I alpha 1 chain | Thermo Fisher | HS00164004_m1 | qPCR primers |

*Appendix 1 Continued on next page*

*Appendix 1 Continued*

| Reagent type (species) or resource | Designation | Source or reference | Identifiers | Additional information |
|---|---|---|---|---|
| Peptide, recombinant protein | Insulin | Sigma-Aldrich | 674889 | |
| Peptide, recombinant protein | Apo-Transferrin human | Sigma-Aldrich | T1147 | |
| Peptide, recombinant protein | 1-Thioglycerol | Sigma-Aldrich | M6145 | |
| Peptide, recombinant protein | Recombinant human FGF-basic (146 a.a.) | PeproTech | 100-18C | |
| Peptide, recombinant protein | FGF-8 | PeproTech | AF-100-25 | |
| Peptide, recombinant protein | Human/mouse recombinant TGF-beta 3 | STEMCELL Technologies | 78131 | |
| Peptide, recombinant protein | Recombinant Human BMP-7 | PeproTech | 120-03P | |
| Commercial assay or kit | Chromium Single-cell 3' Reagent Kits | 10x Genomics | N/A | |
| Chemical compound, drug | DAPI (4',6-Diamidino-2-Phenylindole, Dihydrochloride) | Invitrogen | D1306 | |
| Chemical compound, drug | IMDM, no phenol red | Thermo Fisher | 21056023 | |
| Chemical compound, drug | Ham's F-12 nutrient mix | Thermo Fisher | 11765047 | |
| Chemical compound, drug | Chemically defined lipid concentrate | Thermo Fisher | 11905031 | |
| Chemical compound, drug | Antibiotic-antimycotic solution | Thermo Fisher | 15240096 | |
| Chemical compound, drug | SB 431542 hydrate | Sigma-Aldrich | S4317 | |
| Chemical compound, drug | DMH-1 (CAS 1206711-16-1) | Santa Cruz Biotechnology | Sc-361171 | |
| Chemical compound, drug | CHIR99021 | Biogems | 2520691 | |
| Chemical compound, drug | Y-27632 Dihydrochloride | PeproTech/Biogems | 683093 | |
| Chemical compound, drug | SAG | PeproTech/Biogems | 9128694 | |

*Appendix 1 Continued on next page*

*Appendix 1 Continued*

| Reagent type (species) or resource | Designation | Source or reference | Identifiers | Additional information |
|---|---|---|---|---|
| Chemical compound, drug | LDN193189 | PeproTech/Biogems | 1062443 | |
| Chemical compound, drug | Matrigel | Corning | 354230 | |
| Chemical compound, drug | Geltrex | Thermo Fisher | A1413301 | |
| Chemical compound, drug | DMSO | Sigma-Aldrich | D2650 | |
| Chemical compound, drug | Wnt-C59 | Cayman Chemical | 16644 | |
| Chemical compound, drug | Poly-D-lysine | Thermo Fisher | A3890401 | |
| Software, algorithm | R (v4.1.2) | R Development Core Team | https://www.R-project.org | |
| Software, algorithm | RStudio (v1.4.1103) | RStudio Team (2020). RStudio: Integrated Development for R. RStudio, PBC | https://www.rstudio.com/products/rstudio/ | |
| Software, algorithm | Cell Ranger (v3.0.0) | 10x Genomics | | |
| Software, algorithm | Loupe Cell Browser (v3.0.0) | 10x Genomics | | |
| Software, algorithm | Seurat (v4.1.0) | *Satija et al., 2015*; *Butler et al., 2018*; *Hao et al., 2021*; *Satija et al., 2015*; *Stuart and Satija, 2019* | https://satijalab.org/seurat | |
| Software, algorithm | Monocle (v2.18.0) | *Qiu et al., 2017a*; *Qiu et al., 2017b*; *Trapnell et al., 2014* | http://cole-trapnell-lab.github.io/monocle-release/ | |

