## [Editor Report · eLife Assessment]

The authors established a **useful** syndetome differentiation protocol from human induced pluripotent stem cells, guided by single-cell transcriptomic analysis. Their findings could significantly impact the field, particularly for patients needing tendon cell therapy. However, the evidence presented is currently **incomplete**, as the authors did not yet test the applicability of their protocol across multiple human induced pluripotent stem cell lines.

---

## [Referee Report · Reviewer #1 (Public review)]

Papalamprou et al. established a methodology to differentiate iPSCs to the syndetome stage and validated it by marker gene expression and scRNA-seq analysis. They further found that inhibition of WNT signaling enhanced the homogeneity of the cell population after identifying a group of branching-off cells that overexpressed WNT. Their results will be helpful in developing cell therapy systems for tendon injuries. However, there are several issues to improve the manuscript:

IPA analysis was performed after scRNA-seq. Although it is knowledge-based software with convenient graphic utilities, it is questionable whether an unbiased genome-level analysis was performed. Therefore, it is not convincing if WNT is the only and best signal for the branching-off marker. Perhaps independent approaches, such as GO, pathway, or module analyses, should be performed to validate the findings.

According to the method section, two iPSC lines were used for the study. However, throughout the manuscript, it is not clearly described which line was used for which experiment. Did they show similar efficiency in differentiation and in responses to WNTi? It is also worrisome if using only two lines is the norm in the stem cell field. Please provide a rationale for using only two lines, which will restrict the observation of individual-specific differential responses throughout the study.

How similar are syndetome cells with or without WNTi? It would be interesting to check if there are major DEGs that differentiate these two groups of cells.

Please discuss the improvement of the current study compared to previous ones (e.g., PMID 36203346, 35083031, 35372337).

---

## [Referee Report · Reviewer #2 (Public review)]

Summary:

Dr. Sheyn and colleagues report the step-wise induction of syndetome-like cells from human induced pluripotent stem cells (iPSCs), following a previously published protocol which they adjusted. The progression of the cells through each stage, i.e. presomitic mesoderm (PSM), somitic mesoderm (SM), sclerotome (SCL), and syndetome (SYN) is characterized using FACS, RT-qPCR and immunofluorescence staining (IF). The authors performed also single-cell RNA sequencing (scRNAseq) analysis of their step-wise induced cells and identify signaling pathways which are potentially involved in and possibly necessary for syndetome induction. They then optimized their protocol by simultaneous inhibition of BMP and Wnt signaling pathways, which lead to an increase in syndetome induction while inhibiting off target differentiation into neural lineages.

Strengths:

The authors conducted scRNAseq analysis of each step of their protocol from iPSCs to syndetome-like cells and employed pathway analysis to uncover further insights into somitic mesoderm (SM) and syndetome (SYN) differentiation. They found that BMP inhibition, in conjunction with the inhibition of WNT signaling, plays a role in driving syndetome differentiation. Analyzing their scRNAseq results, they could improve the syndetome induction efficiency of their protocol from 47.6% to 67%-78% while off-target differentiation into neural lineages could be reduced.

Weaknesses:

The authors demonstrated the efficiency of syndetome induction solely by scRNA-seq data analysis before and after pathway inhibition, without using e.g. FACS analysis or immunofluorescence (IF)-staining based assessment. A functional assessment and validation of the induced cells is also completely missing.

---

## [Referee Report · Reviewer #3 (Public review)]

Papalamprou et al sought to fine tune existing tenogenic differentiation protocols to develop a robust multi-step differentiation protocol to induce tendon cells from human GMP-ready iPSCs. In so doing, they found that while existing protocols are capable of driving cells towards a syndetome-like fate, the resultant cultures contain highly heterogeneous cell populations with sub-optimal cell survival. Through single cell transcriptomic analysis they identify WNT signaling as a potential driver of an off-target neural population and show that inhibition of WNT signaling at the later 2 stages of differentiation can be used to promote higher efficiency of generation of syndetome-like cells.

This paper includes a useful paradigm for identifying transcriptional modulators of cell fate during differentiation and a clear example where transcriptional data can be used to guide the chemical modulation of a differentiation protocol to improve cell output. The paper's conclusions are mostly well supported by the data, but the image analysis and discussion need to be improved to strengthen the impact.

The data outlining the differences between the differentiation outcome of the two tested iPSCs is intriguing, but the authors fail to comment on potential differences between the two iPSC lines that could result in drastically different cell outputs from the same differentiation protocol. This is a critically important point, as the majority of the SCX+ cells generated from the 007i cells using their WNTi protocol were found in the FC subpopulation that failed to form from the 83i line under the same protocol. From the analysis of only these 2 cells lines in vitro, it is difficult to assess whether this WNTi protocol can be broadly used across multiple cell lines to generate tenogenic cells. The authors failed to update the text of the manuscript to reflect the potential differences in the two cell lines and the general applicability of their protocol, but rather just include the description of the proposed explanation in the response to reviewer comments. These critical differences in the response to their protocol and their implications for the applications of this proof-of-concept study should be included in the main text.

The authors make claims about changes in protein expression but fail to quantify either fluorescence intensity or percent cell expression from their immunofluorescence analyses to substantiate these claims. The authors state in their response to reviewers that immunofluorescence is qualitative but continue to make quantitative statements such as upregulated or downregulated in both the text and legend describing these images. The authors should either perform the quantification of the IFs, use Western blots for protein quantification of their cell cultures, use Flow Cytometry to count cell numbers, or remove these quantitative words from the description of the images. The image quality and staining specificity continue to be a limitation of this study. These claims are not fully supported by the data as presented as it is unclear whether there is increased expression of tendon markers at the protein level or more cells surviving the protocol.

---

## [Author Response]

The following is the authors’ response to the original reviews.

**Reviewer 1:**
Comment 1: IPA analysis was performed after scRNA-seq. Although it is knowledge-based software with convenient graphic utilities, it is questionable whether an unbiased genome-level analysis was performed. Therefore, it is not convincing if WNT is the only and best signal for the branching-off marker. Perhaps independent approaches, such as GO, pathway, or module analyses, should be performed to validate the finding.

Thanks for your comment. We agree with the reviewer that IPA is a knowledge-based and a hypothesis-driven method. Our hypothesis was that WNT/BMP pathways, among others, are heavily involved in the development of mesenchymal tissues in general and differentiation of tendons specifically. Therefore, we have looked at differentially expressed genes between clusters from a broad array of pathways featured in IPA that could point us towards molecular function that could make a difference. We further corroborated this hypothesis by using WNT inhibitors in subsequent experiments. To address this point, we have supplemented the discussion section with the following remark:

“This study is not without limitations. The IPA network analysis is a knowledge-based and hypothesis driven platform. We have specifically targeted known pathways to be involved in syndetome differentiation. However, WNT signaling stood out with very specific affinity to the off-target populations and we have verified our findings with experiments proving this hypothesis.”

Per the reviewer’s suggestion, we also performed a non-biased GO analysis (Supp. Fig. 6). Multiple pathways were detected in the three clusters of interest (Supp. Fig. 6A-C), including integrin-related and TGFβ-related pathways. However, in these three clusters of interest, WNT signaling was also detected as a prominent pathway. Therefore, we could conclude that it plays a pivotal role in the differentiation process. This hypothesis was later corroborated with WNT inhibitor experiments.

Comment 2: According to the method section, two iPSC lines were used for the study. However, throughout the manuscript, it is not clearly described which line was used for which experiment. Did they show similar efficiency in differentiation and in responses to WNTi? It is also worrisome if using only two lines is the norm in the stem cell field. Please provide a rationale for using only two lines, which will restrict the observation of individual-specific differential responses throughout the study.

Thanks for your comment. This proof-of-concept study is the first investigation that compares data of an in vitro tenogenic induction protocol that has been tested in more than one human iPSC lines. We agree that line-specific phenomena are difficult to interpret and reproduce. Therefore, it is critical to provide data supporting that the findings can be reproduced in more than one line. Some early studies used one line as proof of concept, however now we realize the need to show that the protocol works in at least one additional line.

Here we used the GMP-ready iPSC line CS0007iCTR-n5 for all optimization experiments. This newer low passage feeder-free line was generated from PBMCs and was designated as GMP-ready in the manuscript because it has been derived and cultured using cGMP xeno-free components (mTESR plus medium and rhLaminin-521 matrix substrate instead of Matrigel). We then wanted to confirm the application of the optimized protocol using the reference control line CS83iCTR-22n1 which has already been more widely used by our group1-5 and others.6 This line has been derived from fibroblasts and has been grown and expanded using MatrigelTM and mTESR1, followed by mTESR plus media.

The question of number of lines needed is stage-dependent. In our opinion at the proof-of-concept level, two lines, one of which has been generated in GMP-like conditions is sufficient. Confirmation with multiple lines becomes more pertinent as we move towards scale-up/manufacturing, where considerations regarding robustness and consistency are raised. However, at this stage, it is crucial to understand the developmental processes that are involved in cell differentiation to ensure a more robust protocol can be modified and adapted later. In future studies, as we move towards clinical translation, it is warranted that the approach presented in this work will be further optimized and subsequently evaluated using at least 3 different cell lines that have been generated from various sources.

Comment 3: How similar are syndetome cells with or without WNTi? It would be interesting to check if there are major DEGs that differentiate these two groups of cells.

Thanks for your comment. Single cell RNAseq analysis revealed that treatment with WNTi upregulated tenogenic markers. In SYNWNTi, the expression levels of stage-specific markers COL1A1, COL3A1, SCX, MKX, DCN, BGN, FN1, and TNMD were higher compared to the untreated SYN group, as shown in Figure 5C. Density plots depicted an increase in the number of cells expressing COL1A1, COL3A1, SCX and TNMD in SYNWNTi compared to the SYN group, as illustrated in Figure 5D. Trajectory analysis of the WNTi-treated group revealed the absence of bifurcations observed in the untreated group (Fig. 5E). Therefore, it can be conjured that syndetome cells with and without WNTi are different.

Comment 4: Please discuss the improvement of the current study compared to previous ones (e.g., PMID 36203346 my study, 35083031- Tsutsumi, 35372337- Yoshimoto).

Thanks for your comment. In Papalamprou et al (2023)3, we differentiated iPSCs to mesenchymal stromal-like cells (iMSCs), which were then cultured into a 2D dynamic bioreactor for 7 days. In that study, we examined the impact of simultaneous overexpression of the tendon transcription factor Scleraxis (SCX) using a lentiviral vector and mechanical stimulation on the process of tenogenic differentiation. Following 7 days of uniaxial cyclic loading, we observed notable modifications in the morphology and cytoskeleton organization of iPSC-derived MSCs (iMSCs) overexpressing SCX. Additionally, there was an increase in extracellular matrix (ECM) deposition and alignment, along with upregulation of early and late tendon markers. This proof-of-concept study showed that iPSC-derived MSCs could be a viable cell candidate for cell therapy applications and that mechanical stimulation is contributing to the differentiation of iMSCs towards the tenogenic lineage.

Similarly, Tsutsumi et al7 overexpressed the tendon transcription factor Mohawk (MKX) stably in iPSC-derived MSCs using lentiviral vectors. These cells were then used to seed collagen hydrogels which were mechanically stimulated in a cyclic stretch 3D culture bioreactor for 15 days to create artificial tendon-like tissues, which the authors termed “bio-tendons”. Bio-tendons were then decellularized to remove cellular remnants from the xenogeneic human iPSC-derived cells and were subsequently transplanted in an in vivo Achilles tendon rupture mouse model. The authors reported improved histological and biomechanical properties in the Mkx-bio-tendon mice vs. the GFP-bio-tendon controls, providing another proof-of-concept study in favor of the utilization of iPSC-derived MSCs for tendon cell therapies, while also addressing the immunogenicity of cells of allogeneic/xenogeneic origin. Therefore, the above two studies used tendon transcription factor overexpression and mechanical loading either in 2D or 3D to differentiate MSCs towards the tendon/ligament lineage.

Yoshimoto et al8 optimized a stepwise iPSC to tenocyte induction protocol using a SCX-GFP transgenic mouse iPSC line, by monitoring GFP expression over time. The group performed scRNA-seq to characterize the induction of mesodermal progenitors towards the tenogenic lineage and to shed light into their developmental trajectory. That study unveiled that Retinoic Acid (RA) signaling activation enhanced chondrogenic differentiation, which was in contrast to the study of Kaji et al (2021), which also used a SCX-GFP mouse iPSC line. Kaji et al inhibited TGF and BMP signaling during the process of mesodermal induction and reported that RA signaling eliminated SCX induction entirely and promoted a switch to neural fate. Yoshimoto et al suggested that variations in mesodermal cell identity could be due to the different methods used for mesodermal differentiation. In contrast to the Kaji et al study, Yoshimoto et al opted to stimulate WNT and block the Hedgehog pathway during mesoderm induction. Loh et al (2016) identified the branchpoint from the primitive streak to either the paraxial mesoderm (PSM) or the lateral plate mesoderm (LPM) as the result of two mutually exclusive signaling conditions. Specifically, they reported that induction of PSM was achieved through BMP suppression and WNT stimulation, while the specification of lateral mesoderm was accomplished by BMP stimulation and WNT suppression, all with concurrent TGFβ suppression/FGF stimulation. Lastly, a similar approach towards PSM induction from primitive streak (TGF off/BMP off/ WNT on/FGF on) has been used by many subsequent studies Matsuda et al (2020),9 Wu et al (2021)10 and Nakajima et al (2021).11 The diversity of the above-mentioned approaches points to the plasticity of mesodermal progenitors and the need for additional studies to better understand mesodermal specification and subsequent induction towards sclerotome and syndetome.

In the current study we optimized a stepwise differentiation protocol using xeno-free cGMP ready media and two different cell lines, one of which was cGMP-ready. We used scRNA-seq to characterize the differentiation, which led us to identify off-target cells that were closer to a neural phenotype. We performed pathway analyses and hypothesized that WNT signaling activity might have contributed to the emergence of the off-target cells. To test this, we used a WNT inhibitor (PORCN) to block WNT activity at the SCL stage and at the SYN stage. We found that blockade of WNT signaling at the end of the SM stage and during SCL and SYN induction resulted in a more homogeneous population, while eliminating the neural-like cell cluster. This is the first study that utilized scRNA-seq to shed light into the developmental trajectory of stepwise iPSC to tendon differentiation of human iPSCs and provided a proof-of-concept for the generation of a more homogeneous syndetome population. Further studies are needed to further fine-tune both the process and the final product, as well as elucidate the functionality of iPSC-derived syndetome cells in vitro and in vivo.

**Reviewer 2:**
General concerns: The authors demonstrated the efficiency of syndetome induction solely by scRNA-seq data analysis before and after pathway inhibition, without using e.g. FACS analysis or immunofluorescence (IF)-staining based assessment. A functional assessment and validation of the induced cells is also completely missing.

We appreciate and agree with the reviewer’s critique regarding further analyses of differentiated iPSC-derived syndetome-like cells, including functional assessment of the differentiated cells. Immunofluorescence was used at all timepoints of induction for phenotype confirmation (Fig. 2,4). Flow cytometry for DLL1 was utilized to benchmark efficient differentiation to PSM Loh et al,12 Nakajima et al11. Specifically, DLL1 expression was assessed with flow cytometry after 4 days of induction, and was used to optimize the parameter of initial iPSC aggregate seeding density, which has been previously found to be crucial for in vitro differentiation protocols (Loh et al12). Unfortunately, this parameter is usually not reported although it could be critical to establish protocol replication between different lines.

The function of tendon progenitors is usually reported as response to mechanical cues and the ability to regenerate tendon injuries. In future studies we intend to assess the functionality of the generated syndetome and tendon progenitors and their response to in vitro biomechanical stimulation as previously reported to iMSCSCX+ cells3, 13 and in vivo in a critical tendon defect similarly to what has been previously reported.2

Comment 1: Notably, in Figure 1D, certain PSM markers (TBXT, MSGN1, WNT3A) show higher expression on day 3. If the authors initiate SM induction on day 3 instead of day 4, could this potentially enhance the efficiency of syndetome-like cell induction?

Thanks for your comment. In the current work, we initially optimized differentiation to PSM via expression of DLL1, whose gene expression peaked at d4. We found that this was influenced by the initial iPSC aggregate seeding density. We wanted to generate a homogeneous DLL1+ population which we assessed via gene expression, flow cytometry, IF and scRNA-seq (Fig. 1D, 2C, 3C and Suppl. Fig.1). Given the fact that different lines might display a diverse developmental timeline, we also confirmed reproducibility of the protocol with a second cell line. We appreciate the reviewer’s suggestion to investigate additional protocol iterations, such as the proposed one at the PSM stage, as we move towards a better understanding of key developmental events during in vitro induction.

Comment 2: In the third paragraph of the result section the authors note, "Interestingly, SCX, a prominent tenogenic transcription factor, was significantly downregulated at the SCL stage compared to iPSC, but upregulated during the differentiation from SCL to SYN." Despite this increase, the expression level of SCX in SYN remains lower than that in iPSCs in Fig.1G and Fig.3C. Can the authors provide an explanation for this? Can the authors provide IF data using iPSCs and compare it with in vitro-induced SYN cells? Can the authors provide e.g. additional scRNA-seq data which could support this statement?

Thank you for your comment. In Fig. 1G, SCX expression in SYN was upregulated compared to SCL, however, it was shown to be similar to iPSCs. This suggests a baseline stochastic expression of SCX possibly stemming from spontaneous differentiation of iPSCs in culture (Fig. 3C). Previous research has shown that tenogenic marker gene expression tends to reduce during postnatal tendon maturation Yin et al., 2016b14 Grinstein et al., 2019.15 Yoshimoto et al (2022) utilized a transgenic mouse iPSC-SCX-GFP line to track SCX expression. It was shown that SCX expression peaked after 7d of tenogenic induction and was then decreased at day 14, which marked the end of tenogenic induction. The authors postulated that this pattern of gene expression could either indicate further maturation of tenocytes at subsequent time points, or that the number of non-tenogenic cells increased from T7 to T14.

In the present work, we showed SCX gene expression upregulation in SYN compared to SCL, as well as significant upregulation of TNMD, EGR1, COL1A1 and COL3A1 (Fig.1G). Supp. Fig.8 has been added to show feature plots of SCX and TNMD expression from SCL, SYN and SYNWNTi. The significant upregulation of later markers of tenogenic differentiation suggests that the 21 days of tenogenic induction might have matured the cells. Since gene expression analysis only conveys a snapshot of the transcriptional profile of a cell population, it is likely that we might have missed the peak of SCX upregulation (Supp. Fig. 5). Following treatment with the WNT inhibitor, the SYNWNTi group displayed increased SCX expression (% cells expressing SCX) compared to SYN, which might also be due to a more homogeneous population of syndetome-like cells following treatment with WNTi. In the SYNWNTi group, TNMD was shown to be expressed in the SYN cluster, whereas SCX was mostly found in the cluster that was labelled as fibrocartilage (FC) cluster based on the expression of COL2A1/SOX9/FN1/BGN/COL1A1 markers. Due to the fact that SCX+/SOX9+ progenitor cells are able to give rise to both tendon and cartilage (Sugimoto 2013)16, it could be postulated that this cluster contains tendon progenitors. Interestingly, the FC cluster was not observed in the second iPSC line that we tested, which resulted in a more homogeneous induction to syndetome (78.5% vs. 66.9% SYN cells, Supp. Table 1 & Supp. Fig.3). This slight discrepancy between the two lines and more specifically the presence of the FC cluster only in the 007i line, warrants further investigation. Taken together, these data indicate that the tenogenic induction duration could likely be shortened. Further work to assess the time course of SCX expression over the entire tenogenic induction could be used to further optimize the in vitro induction. For instance, a human edited iPSCSCX-GFP+ line could be generated and used to track SCX expression during the entire induction.

Comment 3: In the fourth paragraph of the result section the authors state, "SM markers (MEOX1, PAX3) and SCL markers (PAX1, PAX9, NKX3.2, SOX9) were upregulated in a stepwise manner." However, the data for MEOX1 and NKX3.2 seems to be missing from Figure 3B-C. The authors should provide this data and/or additional support for their claim.

Thanks for your comment. Feature plots for MEOX1 and NKX3.2 have been added to the Supplemental information (Supp. Fig. 9).

Comment 4: In Figures 2B and 2E, the background of the red channel seems extremely high. Are there better images available, particularly for MEOX1? Given the expected high expression of MEOX1 in SM cells, the authors should observe a strong signal in the nucleus of the stained somitic mesoderm-like cells, but that is not the case in the shown figure. The authors should provide separate channel images instead of merged ones for clarity. The antibody which the authors used might not be specific. Can the authors provide images using an antibody which has been shown to work previously e.g. antibody by ATLAS (Cat#: HPA045214)?

As requested by the reviewer, we have provided separate channels for those images in the Supplement (Supp. Fig. 7). The images show relatively high expression of these markers in SM cells.

Comment 5: In Fig. 2C and Supplementary Fig. 1, the authors present data from immunofluorescence (IF) staining and FACS analysis using a DLL1 antibody. While FACS analysis indicates an efficiency of 96.2% for DLL1+ cells, this was not clearly observed in their IF data. How can the authors explain this discrepancy? Could the authors quantify their IF data and compare it with the corresponding FACS data?

Thanks for your comment. We performed flow cytometric analysis of DLL1 expression to optimize cell seeding density using the 007i line. In the present study, we used IF only in a qualitative manner, that is to confirm protein expression of selected markers. It could be noted that the use of poly-lysine coated coverslips, which are needed for IF, might have slightly altered the density of the cells on the coverslip vs. the plate. Lastly, it cannot be ruled out that the different substrate could have influenced their phenotype differentially through matrix interactions and signaling. On the other hand, flow cytometry by nature is a quantitative and single cell approach, whereas IF staining is qualitative. Therefore, for the purpose of this proof-of-concept work, we tend to trust the quantitative data from the flow cytometry results more than semi-quantitative confirmation achieved through IF staining using coverslips.

Comment 6: In Fig. 2G, PAX9 is expected to be expressed in the nucleus, but the shown IF staining does not appear to be localized to the nucleus. Could the authors provide improved or alternative images to clarify this? The authors should use antibodies shown to work with high specificity as already reported by other groups.

Thanks for your comment. Indeed, the staining seems to be mostly cytoplasmic. We have used antibodies that were previously reported3 and repeated the staining, however, the same results were replicated. We can speculate that this transcription factor has additional role in the iPSC-derived cells and might be traveling to the cytoplasm. Unfortunately, we have no evidence to this phenomenon.

Comment 7: Why did the authors choose to display day 10 data for SYN induction in Fig. 4A? Could they provide information about the endpoint of their culture at day 21?

Thank you for your comment. In Fig. 1G we provided gene expression analyses results for several selected early and later tendon markers for the endpoint of our culture, that is day 21. Following scRNA-seq at each stage of the differentiation (iPSC at d0, PSM at d4, SM at d8, SCL at d11 and the endpoint day 32 for SYN), we performed DEG analysis using the IPA platform. We identified activation of genes associated with the WNT signaling pathway in the off-target clusters. We hypothesized that WNT pathway inhibition might block the formation of unwanted fates and induce a more homogeneous differentiation outcome. We thus tested a WNT inhibitor and compared the inhibitor-treated group with a non-treated group. We then assessed selected neural markers during the course of the inhibitor application. In Fig. 4A we presented gene expression of key selected markers at day 21 using qPCR, which was approximately in the middle of the syndetome induction. Since we observed that the inhibitor downregulated the selected neural markers, we then applied the inhibitor until the endpoint of the initial induction and proceeded to analyze the results using scRNA-seq (Fig. 5). Lastly, it should be acknowledged that this was a proof-of-concept study, and additional optimizations are needed regarding the application of the inhibitor (timing, duration, concentration, etc).

Comment 8: In Supplementary Fig. 5, the authors depicted the expression level of SCX, a SYN marker, which peaked at day 14 and then decreased. By day 21, it reached a level comparable to that of iPSCs. Given this observation, could the authors provide a characterization of the cells at day 21 during SYN induction using IF? What was the rationale behind selecting 21 days for SYN induction? The authors also need to show 'n numbers'; how many times were the experiments repeated independently (independent experiments)?

Thanks for your comment. During the optimization process, we initially used RT-qPCR to track gene expression of selected tenogenic markers using the 007i line. We found that after 21 days of tenogenic induction there was upregulation of the few established tendon markers, that is COL1A1, COL3A1, EGR1 and quite importantly, the more definitive later tendon marker, TNMD. Thus, we decided to proceed with this protocol prior to testing other compounds including the WNT inhibitor WNT-C59. However, as has been discussed in the manuscript, this extended tenogenic induction resulted in cell attrition without the application of the WNT inhibitor. This phenomenon was ameliorated following WNT inhibition. Thus, it could be postulated that the protocol could be further optimized by shortening tenogenic induction to less than 21 days.

The experiments that were conducted to optimize the differentiation process were repeated independently at least n=3 times using qPCR and IF using two lines, that is the 007i and the 83i line as described in the manuscript. The scRNAseq analysis represents a population of cells from in vitro differentiation that originated from the same donor line, therefore it was performed on n=1 sample at each stage. However, the effects of inhibitor application (sample SYNWNTi) were also confirmed using a second cell line (83i), thus a total of n=2 independent samples were analyzed.

Comment 9: Overall the shown immunofluorescence (IF) data does not appear convincing. Could the authors please provide clearer images, including separate channel images, a bright field image, and magnified views of each staining?

Thanks for your comment. The separate channels images were added to the supplemental data (Supp. Fig. 7). We agree with the reviewer regarding the limitations of IF staining, especially with the added confounding factor of using poly-lysine coated coverslips. We would like to point out, that in the current work IF staining is not the main finding or the primary outcome measure, and that it is only used to further support the differentiation by providing a qualitative assessment of protein presence and localization. We describe in this paper our thesis regarding the limitations of IF and the need for more high-throughput unbiased approaches to quantification when using IF staining. For instance, spatial transcriptomics combined with mass cytometry or flow cytometry could be used for a more unbiased approach. Thus, in the present manuscript we based our conclusion on the quantitative gene expression, single cell sequencing and flow cytometry.

Comment 10: As stated by the authors in the manuscript, another research group performed FACS analysis to assess the efficiency of syndetome induction using SCX antibody, and/or quantification of immunofluorescence (IF) with SCX, MKX, COL1A1, or COL2A1 antibodies. Could the authors conduct a comparative analysis of syndetome induction efficiency both before and after protocol optimization, utilizing FACS analysis in conjunction with an SCX reporter line or antibody staining, e.g. quantifying induction efficiency via immunofluorescence (IF) staining with syndetome-specific marker genes?

Thank you for your comment. As discussed in a previous comment, we agree with the reviewer that the generation of a human iPSC-SCX-GFP line would shed light into SCX expression over the entire course of induction. In the current work we used IF as qualitative confirmation of specific marker expression and we showed the presence of SCX, MKX, COL1 and COL3 in SYNWNTi as well as the absence of neuronal markers. As we also pointed it out in the present manuscript, IF can only be considered as a semi-quantitative assessment burdened with several technical limitations as well as operator bias and lower sensitivity and accuracy compared to flow cytometry or scRNA-seq, unless performed in a more unbiased manner. To further clarify this point, firstly, using poly-lysine coated coverslips for IF staining, results in a different substrate environment compared to the Geltrex-coated plates that were used for the induction. Additionally, we noticed that cells grew overconfluent at the edges of the coverslips. This is an important point, since as we have observed in this work, seeding density is critical for the reproducibility of the protocol. It could further be postulated that a different cell substrate stiffness might also have an effect on this process. In our opinion, in this context IF should rather be used qualitatively and a combination of flow cytometry with scRNAseq should be utilized to draw quantitative conclusions such as induction efficiencies of a certain cell type. Since we also observed inconsistencies with the SCX antibodies we tested, the generation of edited human iPSC lines (such as SCX-GFP, MKX-GFP and TNMD-GFP) would be the preferred approach to further explore the efficiency of differentiation.

Comment 11: To enhance the paper's significance, the authors should conduct functional validation experiments and proper assessment of their induced syndetome-like cells. They could perform e.g. xeno-transplantation experiments with syndetome cells into SCID-mice or injury models. They could also assess whether the in vitro induced cells could be applied for in vitro tendon/ligament formation.

Thanks for your comment. For the purpose of this proof-of-concept in vitro study, our primary goal was to initially evaluate a stepwise tenogenic induction protocol using GMP-ready cell lines and chemically defined media. Then, we wanted to utilize the analytical power of scRNA-seq in order to characterize and optimize the protocol, thus focusing on one developmental stage that is not well understood, that of syndetome specification from sclerotome, and hypothesized that by fine-tuning the WNT pathway we would be able to generate a more homogeneous syndetome cell population. We fully agree with the reviewer that the warranted next steps should be to conduct several functional validation experiments, such as in vitro 2D/3D tendon/ligament formation and in vivo transplantation in allogeneic or xenogeneic injury models.

Comment 12: The authors should also compare their scRNA-seq data with actual human embryo data sets, something which could be done given the recent increase in available human embryo scRNA-seq data sets.

This is a great idea and intriguing study. Unfortunately, not all data sets are available at the moment and specifically embryonic and MSK scRNA-seq data is very scarce, although growing. We have no access to data sets from human tendon development, and thus will have to leave this comparison for future studies.

**Reviewer 3:**
Comment 1: The data outlining the differences between the differentiation outcome of the two tested iPSCs is intriguing, but the authors fail to comment on potential differences between the two iPSC lines that could result in drastically different cell outputs from the same differentiation protocol. This is a critically important point, as the majority of the SCX+ cells generated from the 007i cells using their WNTi protocol were found in the FC subpopulation that failed to form from the 83i line under the same protocol. From the analysis of only these 2 cell lines in vitro, it is difficult to assess whether this WNTi protocol can be broadly used to generate tenogenic cells.

Thanks for your comment. This proof-of-concept study is the first investigation that compares data of an in vitro tenogenic induction protocol that has been tested into more than one cell lines. Using unsupervised clustering we identified 11 clusters, which were classified into 6 cell subpopulations. The only observed difference between the two lines was a small subset that was labeled as fibrocartilage (FC), which displayed expression of both tenogenic and chondrogenic markers. This subpopulation was observed in 007i line but not in the 83i line at the end of the SYN induction. Importantly, DEG analysis also showed that it was enriched for SCX. It has been shown that SCX+/SOX9+ progenitors are a distinct multipotent cell group, responsible for the development of SCX−/SOX9+ chondrocytes and SCX+/SOX9− tenocytes/ligamentocytes (Sugimoto 2013)16. As noted in a previous comment (Comment 2 from Reviewer 1), we might have missed SCX upregulation during the 21-day syndetome induction. This can be further supported by Fig. 5E trajectory analysis which shows that this subpopulation (FC) precedes the SYN cell subpopulation. The fact that this subpopulation was present in one line but not the other, might indicate that 83i line resulted in a more mature tendon population. Therefore, we would rather posit that in the case of 83i line, it might not be that the FC subpopulation failed to form, but rather that it was missed in our scRNAseq endpoint analysis which showed that a more homogeneous SYN population was formed (8.7 % in 007i vs. 0.26 % in 83i, Supp. Table 1 & Supp. Fig. 3B). Future studies are warranted to characterize the SYN induction timeline as it pertains to SCX expression followed up by maturation from tenogenic progenitor to tenocytes.

Comment 2: The authors make claims to changes in protein expression but fail to quantify either fluorescence intensity or percent cell expression from their immunofluorescence analyses to substantiate these claims. These claims are not fully supported by the data as presented as it is unclear whether there is increased expression of tendon markers at the protein level or more cells surviving the protocol. Additionally, in images where 3 channels are merged, it would be helpful to show individual channels where genes are shown in similar spectra (ie. Fig 2I SCX/MKX). Furthermore, the current layout and labelling scheme of Figure 4 makes it very difficult to compare conditions between SYN and SYNWNTi protocols.

Thanks for your comment. Protein expression at each stage was verified with immunofluorescence cytochemistry whereby cells were cultured onto poly-lysine coated coverslips, which were then fixed, stained and imaged (Fig. 2). However, prior to WNT inhibitor application, we noticed gradual cell attrition in the cultures at the end of differentiation (Fig. 1B, 2I). The images show qualitative differences with and without the WNT inhibitor. This could be attributed to the heterogeneity of the cell population at SCL stage, which was confirmed by scRNA-seq (Fig. 3A). As it has been discussed previously (Reviewer 2 comments 5 & 9), in the current paper we didn’t provide any IF quantitative analysis because of the qualitative nature of the staining technique. In future work another high-resolution imaging modality will be considered like single cell proteomics and flow cytometry or mass cytometry in order to perform a more unbiased quantitative single cell analysis across different stages and samples. Furthermore, we have added single channel images in the supplemental information.

Comment 3: Individual data points should also be presented for all qPCR experiments (ie. Fig 4A). Biological replicate information is missing from several experiments, particularly the immunofluorescence data, and it is unclear whether the qPCR data was generated from technical or biological replicates.

Thanks for your comment. We have added additional information regarding replicates in each figure legend. We have also changed Fig. 4A.

(1) Glaeser JD, Bao X, Kaneda G, et al. iPSC-neural crest derived cells embedded in 3D printable bio-ink promote cranial bone defect repair. Sci Rep. Nov 4 2022;12(1):18701. https://www.ncbi.nlm.nih.gov/pubmed/36333414

(2) Kaneda G, Chan JL, Castaneda CM, et al. iPSC-derived tenocytes seeded on microgrooved 3D printed scaffolds for Achilles tendon regeneration. J Orthop Res. Oct 2023;41(10):2205-2220. https://www.ncbi.nlm.nih.gov/pubmed/36961351

(3) Papalamprou A, Yu V, Chen A, et al. Directing iPSC differentiation into iTenocytes using combined scleraxis overexpression and cyclic loading. J Orthop Res. Jun 2023;41(6):1148-1161. https://www.ncbi.nlm.nih.gov/pubmed/36203346

(4) Sheyn D, Ben-David S, Tawackoli W, et al. Human iPSCs can be differentiated into notochordal cells that reduce intervertebral disc degeneration in a porcine model. Theranostics. 2019;9(25):7506-7524. https://www.ncbi.nlm.nih.gov/pubmed/31695783

(5) Später T, Kaneda G, Chavez M, et al. Retention of Human iPSC-Derived or Primary Cells Following Xenotransplantation into Rat Immune-Privileged Sites. Bioengineering. 2023;10(9):1049. https://www.mdpi.com/2306-5354/10/9/1049

(6) Sareen D, O'Rourke JG, Meera P, et al. Targeting RNA foci in iPSC-derived motor neurons from ALS patients with a C9ORF72 repeat expansion. Sci Transl Med. Oct 23 2013;5(208):208ra149. https://www.ncbi.nlm.nih.gov/pubmed/24154603

(7) Tsutsumi H, Kurimoto R, Nakamichi R, et al. Generation of a tendon-like tissue from human iPS cells. J Tissue Eng. Jan-Dec 2022;13:20417314221074018. https://www.ncbi.nlm.nih.gov/pubmed/35083031

(8) Yoshimoto Y, Uezumi A, Ikemoto-Uezumi M, et al. Tenogenic Induction From Induced Pluripotent Stem Cells Unveils the Trajectory Towards Tenocyte Differentiation. Front Cell Dev Biol. 2022;10:780038. https://www.ncbi.nlm.nih.gov/pubmed/35372337

(9) Matsuda M, Yamanaka Y, Uemura M, et al. Recapitulating the human segmentation clock with pluripotent stem cells. Nature. Apr 2020;580(7801):124-129. https://www.ncbi.nlm.nih.gov/pubmed/32238941

(10) Wu CL, Dicks A, Steward N, et al. Single cell transcriptomic analysis of human pluripotent stem cell chondrogenesis. Nat Commun. Jan 13 2021;12(1):362. https://www.ncbi.nlm.nih.gov/pubmed/33441552

(11) Nakajima T, Nakahata A, Yamada N, et al. Grafting of iPS cell-derived tenocytes promotes motor function recovery after Achilles tendon rupture. Nat Commun. Aug 18 2021;12(1):5012. https://www.ncbi.nlm.nih.gov/pubmed/34408142

(12) Loh KM, Chen A, Koh PW, et al. Mapping the Pairwise Choices Leading from Pluripotency to Human Bone, Heart, and Other Mesoderm Cell Types. Cell. Jul 14 2016;166(2):451-467. https://www.ncbi.nlm.nih.gov/pubmed/27419872

(13) Yu V, Papalamprou A, Sheyn D. Generation of Induced Pluripotent Stem Cell-Derived iTenocytes via Combined Scleraxis Overexpression and 2D Uniaxial Tension. JoVE. 2024/03/01 2024(205):e65837. https://app.jove.com/65837

(14) Yin Z, Hu JJ, Yang L, et al. Single-cell analysis reveals a nestin(+) tendon stem/progenitor cell population with strong tenogenic potentiality. Sci Adv. Nov 2016;2(11):e1600874. https://www.ncbi.nlm.nih.gov/pubmed/28138519

(15) Grinstein M, Dingwall HL, O'Connor LD, Zou K, Capellini TD, Galloway JL. A distinct transition from cell growth to physiological homeostasis in the tendon. Elife. Sep 19 2019;8. https://www.ncbi.nlm.nih.gov/pubmed/31535975

(16) Sugimoto Y, Takimoto A, Akiyama H, et al. Scx+/Sox9+ progenitors contribute to the establishment of the junction between cartilage and tendon/ligament. Development. Jun 2013;140(11):2280-2288. https://www.ncbi.nlm.nih.gov/pubmed/23615282